# Specific and comprehensive genetic targeting reveals brain-wide distribution and synaptic input patterns of GABAergic axo-axonic interneurons

Ricardo Raudales[1,2†, ‡], Gukhan Kim[1], Sean M Kelly[1,2], Joshua Hatfield[1,3], Wuqiang Guan[1], Shengli Zhao[3], Anirban Paul[1,4], Yongjun Qian[1,3], Bo Li[1,3,5], Z Josh Huang[1,3,5]*

[1]Cold Spring Harbor Laboratory, Cold Spring Harbor, United States; [2]Program in Neurobiology, Stony Brook University, Stony Brook, United States; [3]Department of Neurobiology, Duke University, Durham, United States; [4]Department of Neural and Behavioral Sciences, Penn State College of Medicine, Hershey, United States; [5]Department of Biomedical Engineering, Duke University, Durham, United States

*For correspondence:
Josh.huang@duke.edu

Present address: [†]Department of Psychiatry, Columbia University, New York, United States; [‡]Division of Molecular Therapeutics, New York State Psychiatric Institute, New York, United States

Competing interest: The authors declare that no competing interests exist.

**Abstract** Axo-axonic cells (AACs), also called chandelier cells (ChCs) in the cerebral cortex, are the most distinctive type of GABAergic interneurons described in the neocortex, hippocampus, and basolateral amygdala (BLA). AACs selectively innervate glutamatergic projection neurons (PNs) at their axon initial segment (AIS), thus may exert decisive control over PN spiking and regulate PN functional ensembles. However, the brain-wide distribution, synaptic connectivity, and circuit function of AACs remain poorly understood, largely due to the lack of specific and reliable experimental tools. Here, we have established an intersectional genetic strategy that achieves specific and comprehensive targeting of AACs throughout the mouse brain based on their lineage (*Nkx2.1*) and molecular (*Unc5b*, *Pthlh*) markers. We discovered that AACs are deployed across essentially all the pallium-derived brain structures, including not only the dorsal pallium-derived neocortex and medial pallium-derived hippocampal formation, but also the lateral pallium-derived claustrum–insular complex, and the ventral pallium-derived extended amygdaloid complex and olfactory centers. AACs are also abundant in anterior olfactory nucleus, taenia tecta, and lateral septum. AACs show characteristic variations in density across neocortical areas and layers and across subregions of the hippocampal formation. Neocortical AACs comprise multiple laminar subtypes with distinct dendritic and axonal arborization patterns. Retrograde monosynaptic tracing from AACs across neocortical, hippocampal, and BLA regions reveal shared as well as distinct patterns of synaptic input. Specific and comprehensive targeting of AACs facilitates the study of their developmental genetic program and circuit function across brain structures, providing a ground truth platform for understanding the conservation and variation of a bona fide cell type across brain regions and species.

## eLife assessment

The authors develop a novel genetic strategy for specific and comprehensive labeling of axo-axonic cells, also referred to as chandelier cells, in the mouse brain. The approach and analysis are rigorous such that the data **convincingly** support the key conclusions, including the expanded distribution of axo-axonic cells throughout the brain. This study provides **important** new information about the distribution of a significant neuronal cell type, as well as new tools for future studies. This work will

be of broad interest to neuroscientists who work on the anatomical and functional organization of neural circuits.

## Introduction

The precision of inhibitory control is crucial for regulating information processing and routing in brain circuits (*Roux and Buzsáki, 2015*). In the cerebral cortex, inhibitory signaling is mediated by a large array of GABAergic interneurons (INs), which regulate the input and output of glutamatergic projection neurons (PNs), together forming dynamic ensembles of information processing and relay. Cortical INs comprise highly diverse subpopulations distinguished by their anatomical and physiological properties and gene expression profiles (*Fishell and Kepecs, 2020*; *Gouwens et al., 2020*; *Huang and Paul, 2019*; *Tremblay et al., 2016*). Recent single-cell RNAseq and computational analyses have delineated the hierarchical tree-like organization of cortical GABAergic neurons from subclasses and supertypes down to transcriptomic subtypes (*Callaway et al., 2021*; *Tasic et al., 2018*). Among these, the axo-axonic cells (AACs), or chandelier cells (ChCs), represent the most distinctive and a bona fide type, as they exclusively innervate the axon initial segments (AIS) of pyramidal neurons (PNs) (*Somogyi, 1977*; *Somogyi et al., 1982*). Because AIS are the site of action potential initiation and individual AACs innervate hundreds of PNs in their vicinity, AACs are thought to exert decisive control over spike generation in ensembles of PNs (*Crick et al., 1986*; *Howard et al., 2005*), thereby regulating network operations and information relay. Thus, a more complete understanding of AACs will not only provide key insights into the functional organization of cortical circuits but also facilitate an integrated multi-modal definition of ground truth neuron types, shedding light on the granularity of hierarchical cell-type organization. However, current knowledge on AAC distribution, connectivity, and function remain incomplete, largely due to the lack of reliable experimental tools.

An ideal genetic tool for studying AACs, or any other bona fide neuron type, would have two important features: specificity (targeting only AACs but no other cells) and comprehensiveness or even completeness (capturing if not all AACs). Genetic labeling of AACs was first achieved through fate mapping neural progenitors of the medial ganglionic eminence (MGE) at late embryonic stages using the *Nkx2.1*$^{CreER}$ mouse line (*Taniguchi et al., 2013*). This approach has facilitated analyses of neocortical AAC subtypes (*Wang et al., 2019*), transcriptome profiles (*Paul et al., 2017*), connectivity (*Lu et al., 2017*), synapse development (*Ishino et al., 2017*; *Tai et al., 2019*), and activity-dependent pruning *Wang et al., 2021*; it also enabled a comprehensive fate mapping of AACs across MGE neurogenesis (*Kelly et al., 2019*). However, late embryonic induction in *Nkx2.1*$^{CreER}$ mice is not only cumbersome but also not entirely specific to AACs (*Kelly et al., 2019*), and does not capture all AACs. With the identification of several putative AAC (pACC) molecular markers from scRNAseq datasets (*Paul et al., 2017*; *Tasic et al., 2018*), several gene-knockin driver lines have been generated, which have facilitated studying AACs. Yet, none of these driver lines have been shown to be specific and comprehensive. For example, the *Vipr2*$^{-Cre}$ driver captures a subset of neocortical and basolateral amygdala (BLA) AACs, but is not entirely specific in these areas (*Daigle et al., 2018*; *Nakashima et al., 2022*; *Tasic et al., 2018*). Similarly, our *Unc5b*$^{CreER}$ line enabled adeno-associated virus (AAV) targeting of hippocampal AACs, but also labels endothelial cells when combined with a reporter line (*Dudok et al., 2021*). Despite extensive effort to date, no singularly unique molecular marker for AACs has been identified. Indeed, the same is true for most if not all neuronal subtypes of the mammalian brain. Therefore, specific and comprehensive genetic access to ground truth neuron types remains a broad and major challenge in neural circuit studies.

Here, we leverage lineage origin and molecular markers to establish an intersectional strategy, achieving highly specific and comprehensive targeting of AACs across the mouse brain. We discovered that much beyond the neocortex, hippocampus, and BLA as previously described, AACs are in fact widely distributed across all pallium-derived brain structures and beyond. These include not only the dorsal pallium (dPAL)-derived neocortex and medial pallium (mPAL)-derived hippocampal formation, but also the lateral pallium (lPAL)-derived claustrum–insular complex, the ventral pallium (vPAL)-derived extended amygdaloid complex, and olfactory centers such as piriform cortex. In addition, pAACs are also abundant in anterior olfactory nucleus (AON), taenia tecta (TT) and are also found in the lateral septum and the diencephalic or third-ventricle-derived hypothalamus. The comprehensive labeling of cortical AACs allowed us to quantify their areal, laminar, and axon terminal distribution

**eLife digest** Whether we are memorising facts or reacting to a loud noise, nerve cells in different brain areas must be able to communicate with one another through precise, meaningful signals. Specialized nerve cells known as interneurons act as "traffic lights" to precisely regulate when and where this information flows in neural circuits.

Axo-axonic cells are a rare type of inhibitory interneuron that are thought to be particularly important for controlling the passage of information between different groups of excitatory neurons. This is because they only connect to one key part of their target cell – the axon-initial segment – where the electrical signals needed for brain communication (known as action potentials) are initiated. Since axo-axonic cells are inhibitory interneurons, this connection effectively allows them to 'veto' the generation of these signals at their source.

Although axo-axonic cells have been identified in three brain regions using traditional anatomical methods, there were no 'tags' readily available that can reliably identify them. Therefore, much about these cells remained unknown, including how widespread they are in the mammalian brain. To solve this problem, Raudales et al. investigated which genes are switched on in axo-axonic cells but not in other cells, identifying a unique molecular signature that could be used to mark, record, and manipulate these cells.

Microscopy imaging of brain tissue from mice in which axo-axonic cells had been identified revealed that they are present in many more brain areas than previously thought, including nearly all regions of the broadly defined cerebral cortex and even the hypothalamus, which controls many innate behaviors. Axo-axonic cells were also 'wired up' differently, depending on where they were located; for example, those in brain areas associated with memory and emotions had wider-ranging input connections than other areas.

The finding of Raudales et al. provide, for the first time, a method to directly track and manipulate axo-axonic cells in the brain. Since dysfunction in axo-axonic cells is also associated with neurological disorders like epilepsy and schizophrenia, gaining an insight into their distribution and connectivity could help to develop better treatments for these conditions.

as well as to describe multiple anatomic subtypes. We further extend our strategy to AAV-based targeting and map synaptic input sources to AACs in sensorimotor cortical areas, hippocampal CA1, and BLA, demonstrating feasibility for functional manipulations. Together, these results provide a brain-wide overview of AACs and their subtype variations and input sources in the cerebral cortex. Precise and comprehensive genetic access to AACs sets the stage for understanding many aspects of their role in circuit development and function across the extended brain regions. This establishes a powerful model system to study the conservation and divergence of a ground truth neuron type across different brain regions toward understanding the principles of cell-type organization.

## Results

### Lineage and marker-based intersectional targeting captures AACs across brain regions

The embryonic subpallium generates GABAergic neurons of the telencephalon, with the MGE giving rise to AACs (*Kelly et al., 2019*; *Taniguchi et al., 2013*) while other progenitor domains, such as the caudal ganglionic eminence (*Miyoshi et al., 2015*) or preoptic area (*Gelman et al., 2011*) do not. Within the MGE *Nkx2.1* lineage, AACs are generated in two consecutive waves during both early and late neurogenesis (*Kelly et al., 2019*), extending our initial report of their birth pattern (*Taniguchi et al., 2013*). Therefore, late embryonic induction in the *Nkx2.1*[CreER] line only enriches a subset of neocortical AACs and also labels at least two other IN types *Kelly et al., 2019*; this strategy also misses AACs in the hippocampus, BLA, and potentially other brain areas that are born earlier (*Kelly et al., 2019*). Taking a different approach based on scRNAseq analysis, we identified several pAAC markers (*Paul et al., 2017*) and accordingly generated *Unc5b*[CreER] and *Pthlh*[Flp] driver lines. However, neither by itself is fully specific to AACs: the former also labels endothelial cells and sparse cortical pyramidal neurons (data not shown), and the latter labels a

subset of CGE-derived INs (*Figure 1—figure supplement 3m–p*). To purify AACs from these driver lines, we then designed an intersectional strategy that combines lineage and molecular markers (*Figure 1a*); we tested the intersection of *Nkx2.1*^Flp with *Unc5b*^CreER as well as *Nkx2.1*^CreER with *Pthlh*^Flp. We expected that the constitutive *Nkx2.1*^Flp or Cre line would cover the entire output from MGE, including all AACs, and the intersection with postmitotic markers would specifically capture AACs.

We first combined *Nkx2.1*^Flp and *Unc5b*^CreER driver lines with an Ai65 intersectional reporter allele to generate *Unc5b*^CreER; *Nkx2.1*^Flp; Ai65 mice (*Figure 1b*). Tamoxifen induction (TM) allows dose-dependent titration of AAC labeling density, allowing the characterization of global distribution patterns *en masse* with high dose as well as the evaluation of individual cell morphologies with sparser labeling by lower dose (see methods). Indeed, postnatal high-dose TM induction generated dense and highly characteristic AAC labeling patterns throughout the neocortex, hippocampus, and BLA (*Figure 1d–g*). The characteristic chandelier axon arbor morphology with strings of synaptic boutons (cartridges) along the AIS of pyramidal neurons unequivocally indicated their AAC identity (*Figure 1*, *Figure 1—figure supplements 1–3*), consistent with our previous high-resolution single-cell AAC imaging and reconstructions (*Wang et al., 2019*).

Surprisingly, cell labeling extended much beyond these three previously reported structures that contain AACs. As the AAC identity of labeled cell is validated in most but not all these structures, we tentatively call some of these pAACs. Extensive analysis of these brain regions revealed an overarching pattern: AACs and pAACs are abundantly present in essentially all pallium-derived structures and several additional structures (*Table 1*; *Figure 1c–g*, *Figure 1—figure supplements 1 and 2*, *Video 1*). The dPAL-derived structures include the entire neocortex and entorhinal/ectorhinal cortices. The mPAL-derived structures include the entire hippocampal formation. Beyond these previously known structures, pAACs are abundant in lPAL-derived claustrum, insular, and endopiriform dorsal. In addition, the vPAL-derived structures include most of the amygdaloid complex (cortical amygdala, basolateral and basomedial amygdala, and medial amygdala) and extended amygdala (bed nucleus of stria terminalis). Most surprisingly, AACs are highly abundant in a set of olfactory centers including piriform cortex, AON, and TT. Among these, the piriform derives from the vPAL while AON and tenia tecta dorsal (TTd) derive from septum neuroepithelium. Finally, pAACs are found in lateral septum also derived from septum neuroepithelium (*Magno et al., 2022*) and parts of the hypothalamus (derived largely from neuroepithelium of the third ventricle).

Consistent with these findings, the *Pthlh*^Flp; *Nkx2.1*^Cre; Ai65 mice were found to have an overall similar pattern of pAAC distribution across different brain regions, with several differences (*Figure 1—figure supplement 3*). First, this intersection also labeled a set of layer 4 INs in the somatosensory cortex (*Figure 1—figure supplement 3f*). Second, *Pthlh;Nkx2.1* yielded slightly higher cell targeting in retrosplenial cortex (*Figure 1—figure supplement 3h, i*). Third, *Pthlh;Nkx2.1* labeled fewer cells in the AON and BST. As an additional application of the *Pthlh*^Flp line, we demonstrate that when combined with an inducible *Nkx2.1*^CreER driver (*Taniguchi et al., 2013*) the *Pthlh*^Flp; *Nkx2.1*^CreER; Ai65 mice led to exclusive fate mapping of AACs (*Figure 1—figure supplement 3j–l*). The overall similarity between *Unc5b/Nkx2.1* and *Pthlh/Nkx2.1* patterns (except in layer 4 somatosensory cortex and striatum), and high density of cells captured, provided independent evidence that these intersectional approaches may have captured most if not all AACs in the brain.

In addition to targeting AACs, the *Pthlh*^Flp is a useful tool for at least two additional cell types. First, consistent with its expression in a set of CGE-derived INs (including *Vip*-expressing cells), intersection with the *Vip*^Cre driver line, which alone expresses in other cell types, was highly specific for an IN subtype with striking vertical morphology that may represent IN-selective cells (*Figure 1—figure supplement 3m-p*). Second, *Pthlh/Nkx2.1* intersection yielded dense labeling of striatal INs (i.e. striatal neurons without axons in SNr/GPi) (*Figure 1—figure supplement 3f–h*), consistent with known *Pthlh* expression in striatal *Pvalb* INs (*Muñoz-Manchado et al., 2018*).

In sum, our lineage and marker intersections combining *Nkx2.1*^Flp, *Nkx2.1*^Cre, or *Nkx2.1*^CreER with *Unc5b*^CreER or *Pthlh*^Flp appear to have achieved unprecedented specificity and comprehensiveness of AAC targeting. To validate this result, in the following sections, we present more detailed and quantitative descriptions of AACs and pAACs across brain structures.

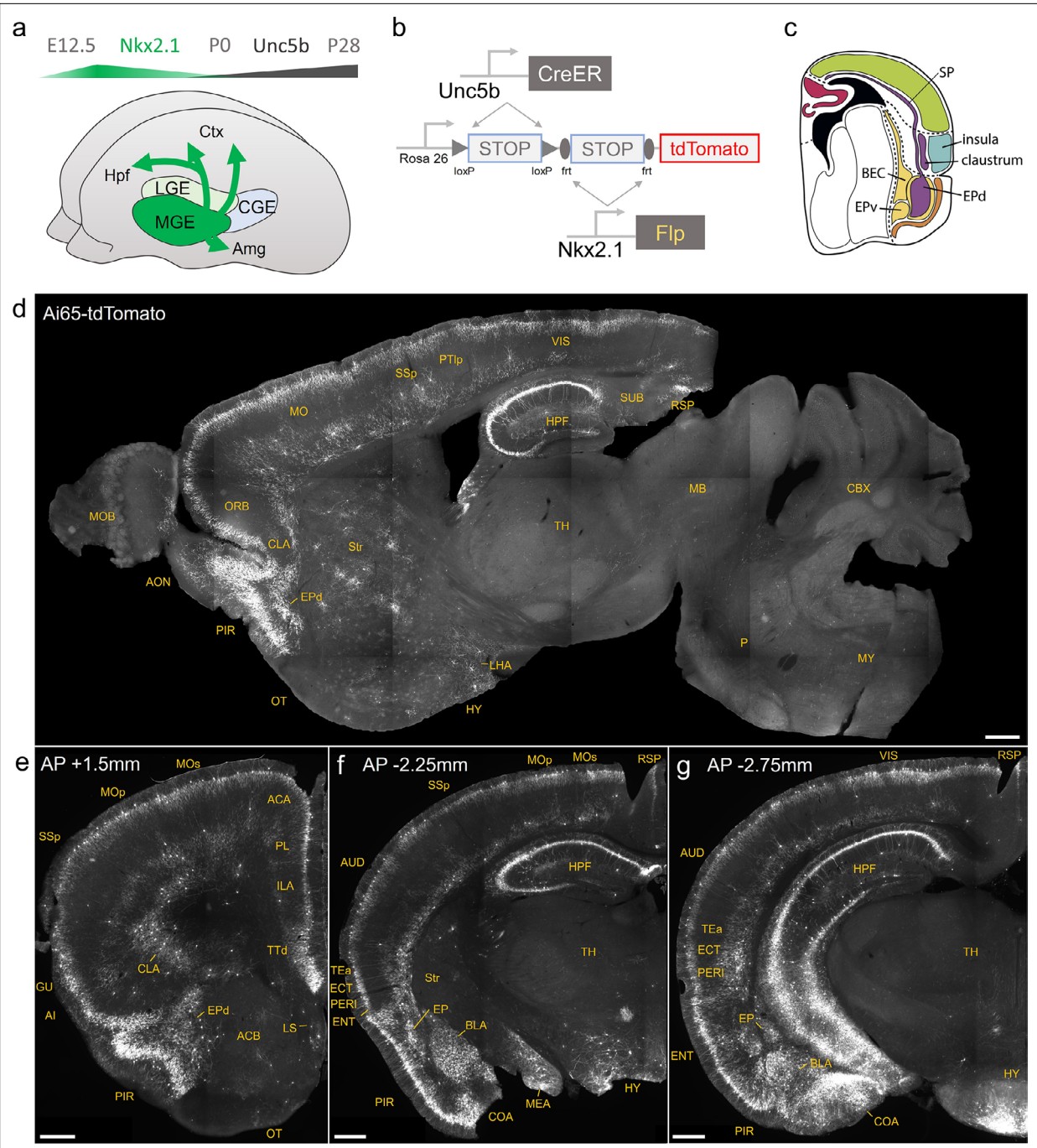

**Figure 1.** Developmental genetic intersectional targeting reveals brain-wide axo-axonic cell (AAC) distribution patterns. Schematic of lineage (*Nkx2.1*) and marker (*Unc5b*) intersection for pan-AAC labeling. (**a**) Embryonic *Nkx2.1* expression defines medial ganglionic eminence (MGE) interneuron identity while subsequent postnatal *Unc5b* expression restricts interneurons to AACs. Height of green or gray shade represents relative expression levels. (**b**) Configuration of triple allele intersectional labeling, combining *Unc5b*^CreER and *Nkx2.1*^Flp drivers and the Cre-AND-Flp Ai65 reporter. Embryonic Flp expression removes the frt-flanked STOP cassette and postnatal *Cre*ER induction removes the loxp-flanked STOP, thereby activating constitutive tdTomato expression. (**c**) Schematic showing that MGE-derived AACs migrate and populate all four pallial-derived brain structures depicted in color: medial pallium (red), dorsal pallium (green), lateral pallium (violet and blue), and ventral pallium (yellow and orange). (**d**) Representative midsagittal section showing AAC labeling in cerebral cortex, hippocampal formation (HPF), and olfactory centers such as piriform cortex (PIR) and anterior olfactory nucleus (AON). Note sparse labeling in lateral hypothalamus (LH) and striatum (str). (**e-g**) Representative coronal sections at specified anterior–posterior coordinates (from Bregma) showing dense AAC labeling in cerebral cortex (e–g), HPF (f, g), claustrum (CLA, e), endopiriform (EP, f, g), taenia tecta (TTd, e), lateral septum (LS, e), basolateral amygdala (BLA, f, g), cortical amygdala (COA, f, g), medial amygdala (MeA, f), and hypothalamus (HYP, g). Scale

*Figure 1 continued on next page*

*Figure 1 continued*

bars, 500 µm. Abbreviations for anatomical structures are listed in ***Supplementary file 1***. All images showing Ai65-tdTomato were immunostained for signal amplification.

The online version of this article includes the following figure supplement(s) for figure 1:

**Figure supplement 1.** Dense labeling of axo-axonic cells (AACs) across brain regions in a *Unc5b*^CreER; *Nkx2.1*^Flp; Ai65 mouse.

**Figure supplement 2.** Additional sagittal sections of *Unc5b*^CreER; *Nkx2.1*^Flp; Ai65 mice highlighting dense putative AACs (pAACs) in medial, lateral, and ventral pallium-derived structures.

**Figure supplement 3.** Intersectional targeting with *Pthlh*^Flp and *Nkx2.1*^Cre, *Nkx2.1*^CreER or *Vip*^Cre to label specific subsets of axo-axonic cells (AACs).

## Areal, laminar, and subtype distribution of AACs across the neocortex

Previous studies have identified ChCs in multiple neocortical areas and cortical layers with the observation of several morphological variants (***Kelly et al., 2019***; ***Somogyi et al., 1982***; ***Taniguchi et al., 2013***; ***Wang et al., 2019***), but the global and quantitative distribution of ChC across the cerebral cortex and cortical layers is unknown. To first validate the specificity of our method to ChCs, we used single low-dose TM induction to sparsely label individual cells across cortical areas (***Figure 2—figure supplement 1***). Combined with immunolabeling of AnkyrinG-stained pyramidal neuron (PN) AIS, we showed that the vast majority of RFP⁺ cells (97.3%, *n* = 110 cells from 6 mice) had axonal cartridge structure with synaptic boutons along PN AIS, validating their ChC identity (***Figure 2—figure supplement 1***).

To systematically map and quantify ChC distribution across the entire cerebral cortex, we then used high-dose TM induction in *Unc5b*^CreER; *Nkx2.1*^Flp; Ai65 mice to densely label ChCs and performed serial two-photon tomography (STP) across the entire mouse brain at 50 µm intervals (***Video 1***). The

**Table 1.** Brain-wide distribution pattern of putative axo-axonic cells (AACs).

| Developmental origin | Brain structure | Evolutionary term | Validation |
|---|---|---|---|
| Dorsal pallium | Neocortex all cortical areas entorhinal, perirhinal | Isocortex | Yes |
| Medial pallium | Hippocampal formation dentate, CA1, CA2, CA3 subiculum | Archicortex | Yes |
| Lateral pallium | Claustrum insular endopirifom dorsal | | Not yet |
| Ventral pallium | Amygdaloid complex<br><br>• BLA<br>• CoA<br>• MEA<br>• BMA<br><br>extended amygdala<br><br>• CeA (no AACs found)<br>• BNST (putative cases only)<br><br>olfactory cortex<br><br>• Piriform<br>• Endopiriform ventral [no] | Paleocortex | Yes |
| Septal neuroepithelium | • Lateral and medial septum<br>• Tania tecta<br>• AON | | Yes |
| Hypothalamo-telencephalic prosomere h1/h2 | Hypothalamus<br><br>• Arcuate nucleus<br>• Periventricular hypothalamic nucleus | | Not yet |

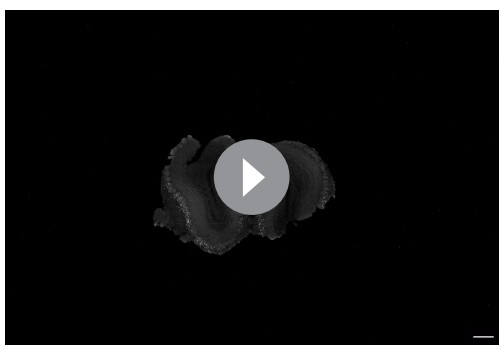

**Video 1.** Whole-brain serial two-photon (STP) imaging and reconstruction of *Unc5b*<sup>CreER</sup>; *Nkx2.1*<sup>Flp</sup>; Ai65. Serially reconstructed coronal images of representative dense-labeled *Unc5b*<sup>CreER</sup>; *Nkx2.1*<sup>Flp</sup>; Ai65 brain imaged by STP microscopy at 50 µm intervals. Scale bar, 500 µm.

https://elifesciences.org/articles/93481/figures#video1

whole-brain STP datasets were then registered to the Allen CCFv3 atlas, followed by segmentation of anatomical regions and an automated cell detection algorithm was applied to identify and quantify labeled cells in each region (*Ragan et al., 2012*; *Kim et al., 2017*; *Matho et al., 2021*). ChCs were detected across all areas of cerebral cortex (*Figure 2a*; *Video 1*). Normalized cell density analysis revealed area-specific patterns (*Figure 2b*). The highest densities were observed in ventrolateral areas such as PERI, ECT, TEA, and AI as well as ventromedially in infralimbic area (ILA). There was a slight decline in ChC density very rostrally in FRP or caudally in RSP.

We further mapped ChC distribution across cortical layers in several sensory and motor areas (*Figure 2c, d*). The majority of ChC somas were located in L2/3, though this was area specific, with normalized density ranging from the lowest in SSs (65.5 ± 0.6%) to highest in VISp (94.7 ± 1.0%). An analysis of relative cortical depth (*Figure 2e*) revealed that ~80% of ChCs occupied a narrow band just below the L1/2 border, though this varied by anterior–posterior location. Nevertheless, ChCs were found at nearly all cortical depths, with a sizable proportion forming a second, less dense band above the white matter tract. This largely 'bilaminar' pattern was highly evident in certain areas such as the SSp (*Figure 2f*).

To more systematically sample the morphological variations of ChCs, we used low-dose TM induction to resolve individual cells across areas (*Figure 2—figure supplement 1*). We identified all the previously described laminar and morphological subtypes (*Wang et al., 2019*), including supragranular (L2/3) (*Figure 2g*), infragranular (L5/6) (*Figure 2h*), and inverted (axon arbors extend above cell soma). Notably, we found rather frequent bi- and tri-laminar subtypes – L2/3 ChCs extending their axon arbors not only locally but further down to another one or two deep layers (*Figure 2i*). These translaminar ChCs may coordinate PN ensembles between cortical layers. The abundance of translaminar ChCs varied among cortical areas, for example with high abundance in auditory cortex (*Figure 2j*).

In addition to soma position and cell morphology, we sought to map the distribution of ChC axon terminals (i.e. cartridges) throughout the mouse cortex. For this purpose, we used a Cre/Flp-dependent reporter line expressing a synaptophysin–EGFP fusion protein and cytoplasmic tdTomato (*Niederkofler et al., 2016*; *Figure 2k–l*). Following high-dose induction and dense labeling, we performed whole brain STP imaging and calculated cartridge density across ARAv3 areas (see Methods). ChC cartridges exhibited a wide range of densities across areas and layers (*Figure 2m*), with overall similarity to ChC soma densities in those areas and layers (*Figure 2b*). Plotting the relative laminar depth for ChC cartridges in each area (*Figure 2n*) revealed characteristic patterns of population-level laminar innervation onto PNs.

## AACs in the hippocampus

Decades of studies of hippocampal AACs using traditional anatomical and physiological methods have implicated their role in network oscillations, including sharp waves and epilepsy (*Fitzgerald et al., 2013*; *Klausberger et al., 2003*; *Viney et al., 2013*) but the quantitative distribution of AACs across hippocampal formation has remained unknown, and reliable genetic access to hippocampal AACs is not well established. We applied *Unc5b/Nkx2.1* intersectional targeting and STP to map AAC distribution throughout hippocampal compartments. Single-cell labeling by low-dose tamoxifen and immunohistochemistry confirmed that the vast majority (98.2%, *n* = 56 cells, 6 mice) were AACs, characterized by vertically oriented cross laminar apical dendrites in CA1–CA3 and axon terminals innervating PN AIS (*Figure 3a, b, d*). Dense labeling with high-dose TM followed by cell density analysis revealed that AAC density was, surprisingly, highest in CA2 and lowest in dentate gyrus (*Figure 3c*), with density in CA1 and CA3 approximately equal. Only a very sparse population of AAC was labeled

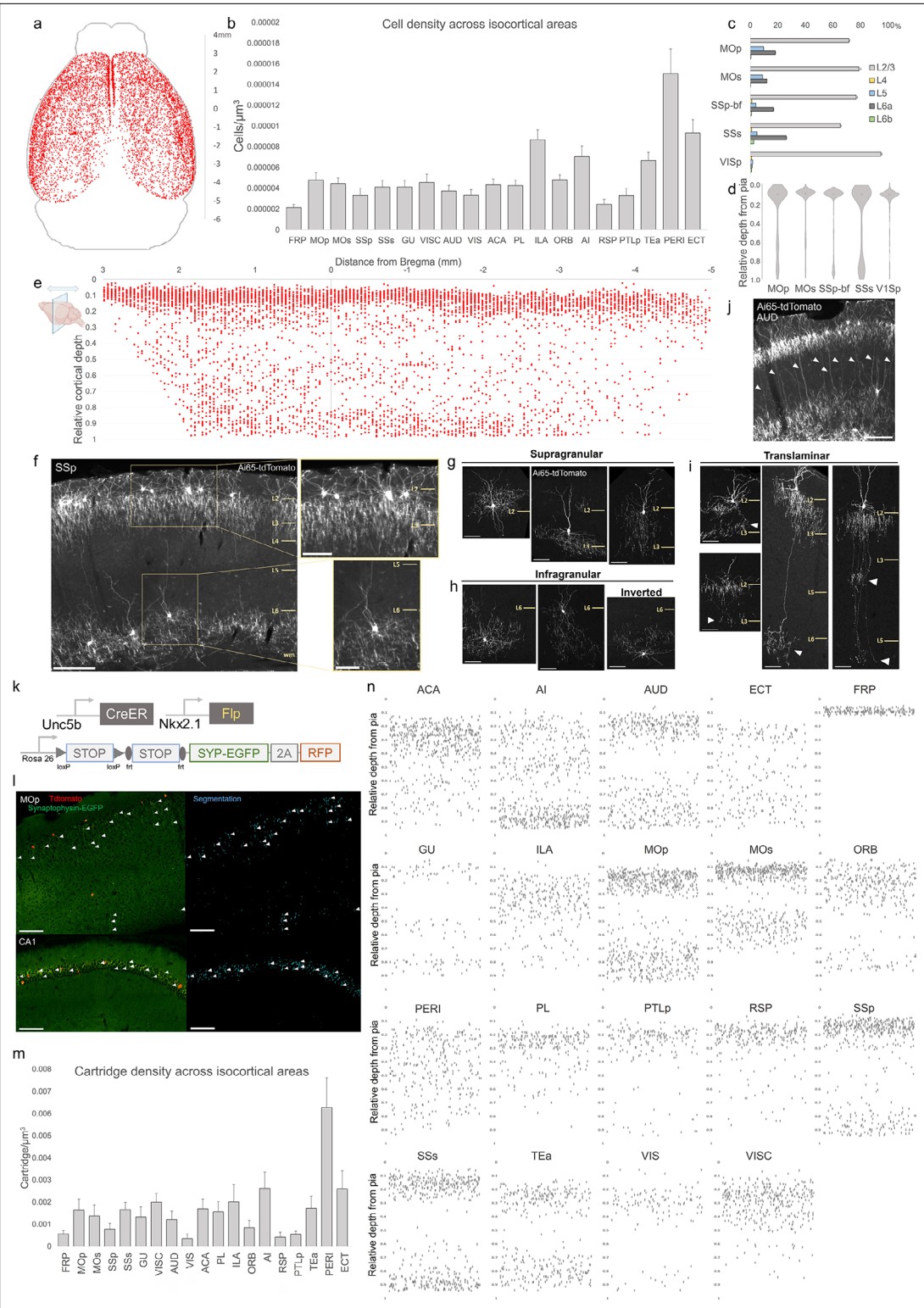

**Figure 2.** Characterization of chandelier cells (ChCs, i.e. cortical axo-axonic cells [AACs]) in cerebral cortex. (**a**) Representative two-dimensional (2D) stereotactic plot of cortical ChC distribution in *Unc5b; Nkx2.1*. Dots represent individual red fluorescent (RFP) labeled cells. Anterior–posterior distance from Bregma (vertical scale on right). (**b**) Normalized ChC cell density (cells/um³) following registration to ARAv3 isocortical areas. For b and c, data are mean ± standard error of the mean (SEM). (**c**) Comparative bar plot showing relative proportion in each cortical layer in sensorimotor cortices. (**d**) Violin plots of ChC cell density proportion along pia-to-white matter cortical depth in each sensorimotor cortical area. Median is used for each violin plot. (**e**) Scatterplot of ChC distribution along pia-to-white matter depth (vertical axis) and anterior–posterior distance from Bregma (horizontal

*Figure 2 continued on next page*

*Figure 2 continued*

axis, in mm); red dots represent individual cells. While a vast majority (~80%) of ChCs occupy a band near the layer I/II border, ChCs in other layers account for ~20% of the total cortical population and vary in depth distribution by anterior–posterior cortical location. (**f**) Representative ChC laminar distribution pattern, showing largely separated supragranular (top inset) and infragranular (bottom inset) laminar subtypes. Scale bars, 200 µm and insets, 100 µm. (**g–i**) Multiple ChC laminar subtypes revealed by single-cell labeling using low-dose tamoxifen induction. These include supragranular subtypes (**g**), infragranular and inverted subtypes (**h**), and translaminar L2/3 subtypes that extend long vertical axons (white arrowheads) to deep layers (**i**). White arrowheads highlight axonal plexi distant from soma in translaminar types. Scale bars, 65 µm. (**j**) Example of regional ChC morphological diversity in auditory cortex, with relatively high proportion of translaminar subtypes. Scale bar, 200 µm. (**k–n**) ChC axon terminal (i.e. 'cartridge') labeling and mapping using intersectional synaptophysin-EGFP/cytoplasmic-tdtomato reporter mouse line. Diagram of intersectional genetic labeling of ChC cartridges (EGFP) and soma (tdTomato). Pixel classifier trained to segment EGFP-expressing ChC cartridges in cortex and CA1 hippocampus. Native EGFP fluorescence was imaged using serial two-photon (STP) microscopy. White arrowheads indicate detected EGFP-labeled cartridges. Scale bars, 100 µm. Density of ChC cartridges (cartridges/µm³) registered to ARAv3. Representative relative depth plots (from pia) of individual detected cartridges at a single AP coordinate for each cortical area. Refer to for ARAv3 area label abbreviations. Note reduced deep layer cartridges in FRP, RSP, and V1.

The online version of this article includes the following source data and figure supplement(s) for figure 2:

**Source data 1.** Mean, SEM and raw depths for data points in *Figure 2b-e, m*.

**Figure supplement 1.** Anatomical diversity of chandelier cells (ChCs) across neocortical areas.

in dentate gyrus. Overall AAC density in CA2 was 3- to 4-fold greater than CA1/CA3 and 13-fold greater than dentate gyrus (DG) (*Figure 3c*). As the density of PNs across hippocampal areas is overall similar, this result suggests substantially different AAC–PN innervation ratio and that AACs may exert much denser and likely stronger control of CA2 PNs than CA1/CA3 PNs or DG granule cells.

## AACs in anterior insular, claustrum, and dorsal endopiriform nuclei

According to the updated tetrapartite model of pallial/cortical development (*Figure 1c*), the lPAL gives rise to the claustro-insular complex as well as the dorsal endopiriform nuclei (EPd) (*Puelles, 2017*). Our brain-wide imaging of *Unc5b*^CreER^; *Nkx2.1*^Flp^; Ai65 mice revealed pAACs deployed across these three regions (*Figure 4a*). Unlike cortical and hippocampal AACs whose axon arbors exhibit highly characteristic laminar pattern, reflecting the organization of their postsynaptic targets (i.e. AIS of PNs), the axon arbors of RFP cells in AI, CLA, EPd do not show clear laminar patterns and exhibit less uniformity and are more multipolar in morphology (*Figure 4b, c*), making it more difficult to ascertain their identity. Nevertheless, strings of synaptic boutons were frequently identified that co-localized with AnkG-labeled AIS, suggesting these as pAACs. Quantification from dense labeling revealed higher pAAC density in EPd compared to claustrum and EPv, the latter of which structure is derived from vPAL (*Figure 4d*).

## AACs in the amygdaloid complex and extended amygdala

The amygdaloid complex includes over a dozen nuclei and can be segregated into five groups (*Beyeler and Dabrowska, 2020*): (1) the BLA divided into a dorsal section (lateral amygdala, LA) and basal section (basal amygdala, BA), (2) the basomedial amygdala (BMA), (3) the central amygdala (CeA) further splits into medial, lateral, and central sections (CeM, CeL, and CeC), (4) the medial amygdala (MeA), and (5) the cortical amygdala (CoA). Furthermore, the CeM extends rostrally and medially, thereby including the bed nucleus of stria terminalis (BST) to form the extended amygdala. Based on developmental origin, connectivity, and gene expression, these groups can be assigned into two large categories: pallium-derived cortex-like structures and subpallium-derived basal ganglia-like structures. The former includes BLA, CoA, BMA, and MeA, while the latter includes CeA and BST. Within the amygdala nuclei, PNs are exclusively glutamatergic in BLA, CoA, BMA, exclusively GABAergic in CeA, and predominantly GABAergic in MeA and BST. In rodents, there is also a population of glutamatergic pyramidal neurons (GLU PNs, derived from third ventricle neuroepithelium) that populates the BST, MeA, and hypothalamus (*García-Moreno et al., 2010*; *Huilgol and Tole, 2016*).

In this context, we found AACs or pAACs in all the amygdala nuclei containing GLU PNs, that is except CeA (*Figure 4i–h*). Sparse labeling and immunohistochemical co-staining with AnkG confirmed that the labeled cells in these compartments were nearly exclusively AIS-targeting (97.2%, n = 36 cells, 6 mice). Morphologically, AACs in the amygdala tended to be similar to those in the claustrum, exhibiting multipolar dendrites and multipolar axon terminal branches (in contrast to axon arbor of cortical AACs that typically extend either below or above the soma) (*Figure 4f*). The density of AACs in amygdala appears

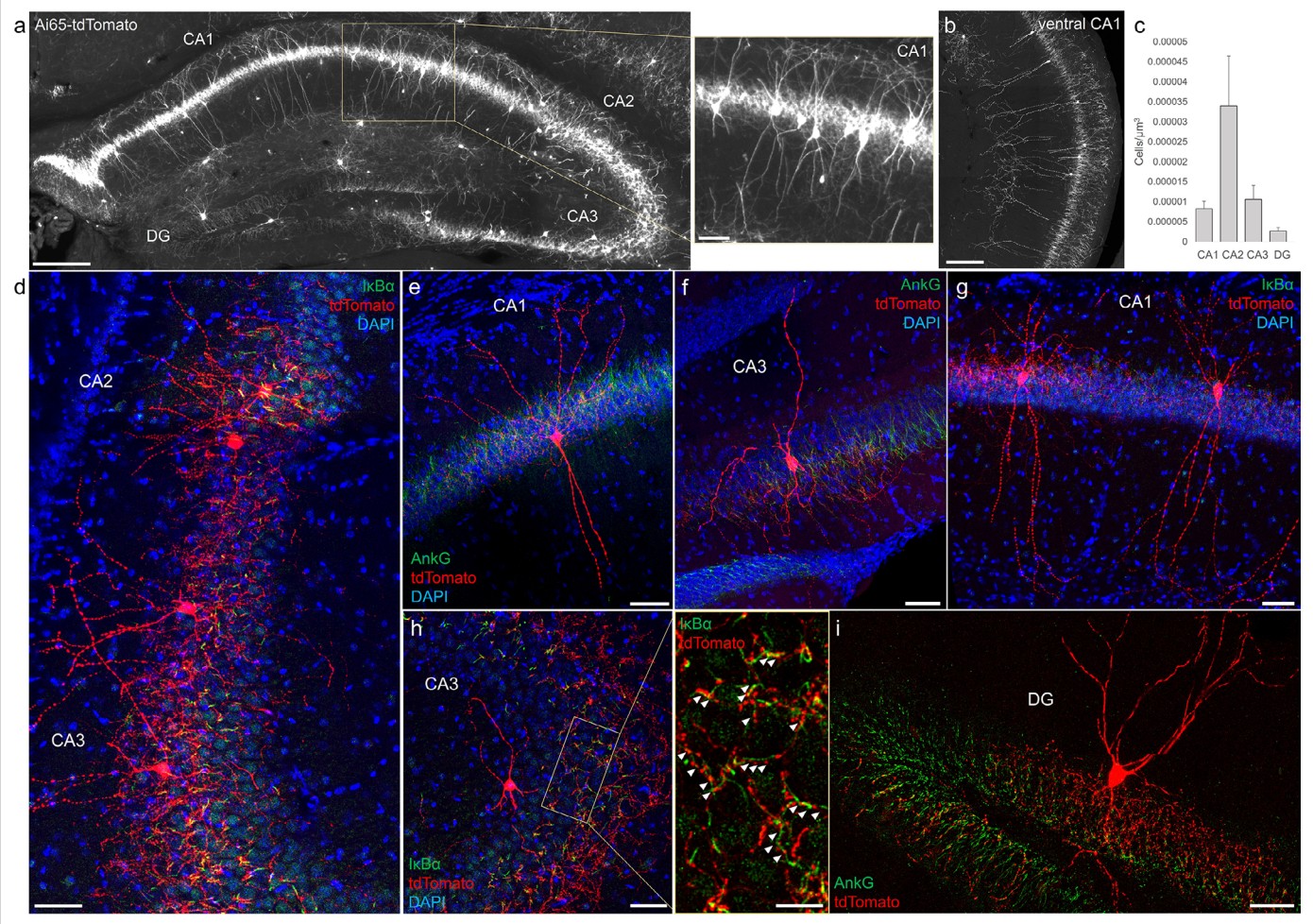

**Figure 3.** Distribution and validation of axo-axonic cells (AACs) in hippocampus. (**a**) Population labeling of AACs throughout hippocampal compartments by *Unc5b; Nkx2.1* intersection. Note the highly stereotyped banded distribution of AACs in CA1–CA3, with somata in the stratum pyramidale and vertically oriented dendrites extending basally to the stratum oriens and apically to the striatum lacunosum-moleculare. AACs are much sparser in dentate gyrus, with individual cells having elaborate and extended horizontal axon arbor. Scale bars, 200 and 50 µm for high-magnification inset. (**b**) AACs in ventral CA1 have overall similar morphology as in dorsal CA1, though with longer basal–apical dendrites matching the ventral CA1 anatomy. (**c**) Normalized AAC cell density (cells/µm³) across hippocampal compartments. CA2 has 3- to 4-fold higher AAC cell density compared to CA1 and CA3 and 12-fold higher compared to DG. (**d–i**) Immunohistochemistry validation with the axon initial segment (AIS) markers AnkG or I κ Bα (green) and sparse-labeled RFP cells confirm axo-axonic targeting by the intersectional strategy. Examples of hippocampal AACs in CA1 (**e, g**), CA2 (**d**) and CA3 (**d, f, h**). In contrast, AACs in DG have much wider axon arbor in the stratum granulosum, potentially innervating many more granule cell AIS compared to in other hippocampal compartments (**i**). High-magnification inset: white arrowheads indicate segments of I κ Bα and RFP apposition. Scale bars, 50 and 20 µm for high-magnification inset. For b, data are mean ± standard error of the mean (SEM).

The online version of this article includes the following source data for figure 3:

**Source data 1.** Mean and SEM for data points in *Figure 3c*.

---

to correlate with the known abundance of GLU PNs: densities in BLA, CoA, and BMA were much higher than those in MeA and BST. Among the pallium-derived amygdala subdivisions (LA, BLA, BMA, and PA), AACs were present at roughly similar densities (*Figure 4i*), while among striatum-like amygdala nuclei they were largely concentrated in MeA (*Figure 4j*). Our results indicate that AACs are deployed to all amygdala nuclei containing GLU PNs. Within the largely GABAergic structures such as MeA and BST, we hypothesize that pAACs might innervate the minor glutamatergic PNs in these structures.

## AACs in olfactory centers: piriform cortex, AON, and TTd

A major novel finding is the surprising abundance of AACs across multiple olfactory centers. Derived from the vPAL, the piriform cortex had been reported to contain axo-axonic synapses by co-localization

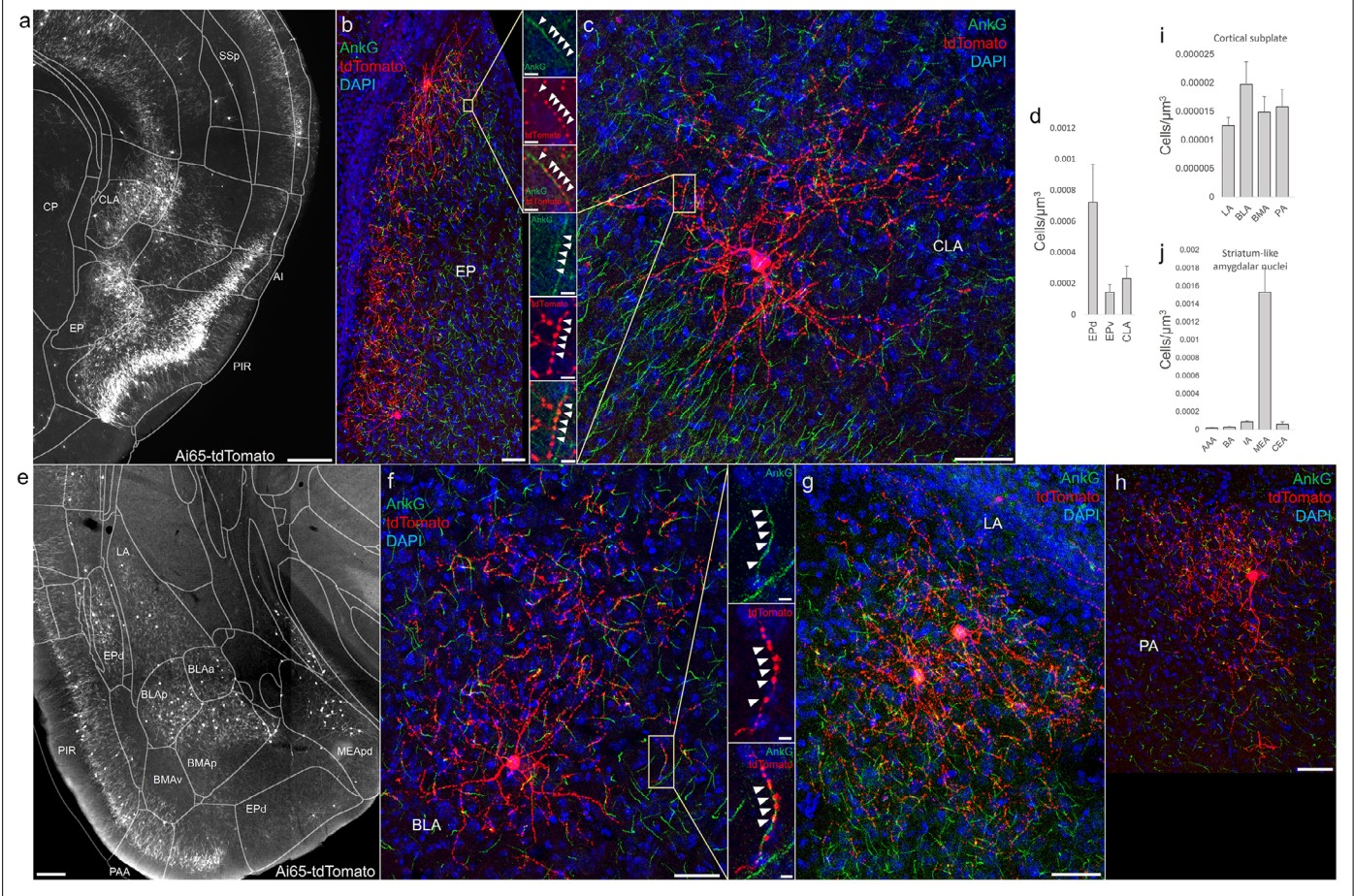

**Figure 4.** Distribution of axo-axonic cells (AACs) across lateral pallium (lPAL)- and ventral pallium (vPAL)-derived structures. (**a**) Dense-labeled RFP cells in claustrum, endopiriform nucleus, and insula. Note AACs in insula follow the banded laminar pattern of AACs similar to cortical upper layers and piriform, while those in claustrum and endopiriform are more dispersed in distribution. Sparse labeling of single-cell morphologies in endopiriform nucleus (**b**) and claustrum (**c**) showing elaborate dendritic and axonal branching structure. High-magnification insets: white arrowheads indicate segments of AnkG and RFP apposition. Scale bars, 50 and 5 µm for single-cell examples and insets, respectively. (**d**) Normalized AAC cell density (cells/µm³) across lateral pallial structures. pAACs are approximately fivefold more prevalent in dorsal EP compared to ventral EP. Data are mean ± standard error of the mean (SEM). (**e**) Dense labeling throughout vPall-derived amygdalar nuclei and piriform cortex. Note there is a degree of compartmentalization, with fewer cells in the ventral-most portions of EPd and the ventral basomedial amygdala (BMA). The laminar pattern of AAC distribution in piriform and Piriform-amygdalar area (PAA) is similar to other three-layered allocortices. (**f–h**) Immunohistochemistry of single-cell labeling shows the multipolar morphologies characteristic of amygdalar AACs. High-magnification inset: white arrowheads indicate segments of AnkG and RFP apposition. Scale bars, 500 µm for grayscale panels, 50 and 5 µm for single-cell examples and insets. Normalized AAC cell density (cells/µm³) across cortical subplate (**i**) and striatum-like (**j**) amygdala. Data are mean ± standard error of the mean (SEM). All images showing Ai65-tdTomato were immunostained for signal amplification, except for a and e which were native fluorescence.

The online version of this article includes the following source data for figure 4:

**Source data 1.** Mean and SEM for data points in *Figure 4d, i, j*.

of the inhibitory presynaptic marker GAT1 and AnkG (*Wang and Sun, 2012*). Our dense labeling in *Unc5b*^CreER; *Nkx2.1*^Flp; Ai65 mice revealed a striking abundance of AACs that span the anterior-medial extent of the piriform in a single layer (*Figure 5a–c*). Sparse-labeled single AACs exhibited characteristic morphologies analogous to L2/3 AACs of the neocortex, with extended branching of apical dendrites to L1 and dense axon arbors below the cell body forming cartridges along PN AIS (*Figure 5d–j*). The single-layer distribution pattern of piriform AACs continued dorsally to the insular cortex and ventrally to piriform-amygdala area (PAA) and COA (*Figure 5c*).

Furthermore, we found that AACs are deployed to two other olfactory centers that derive from outside of the four pallial domains: namely the AON and TT (*Figure 6a, b*). Though exhibiting a range of different morphologies, AACs in all three areas resemble those of L2/3 neocortex, with apical

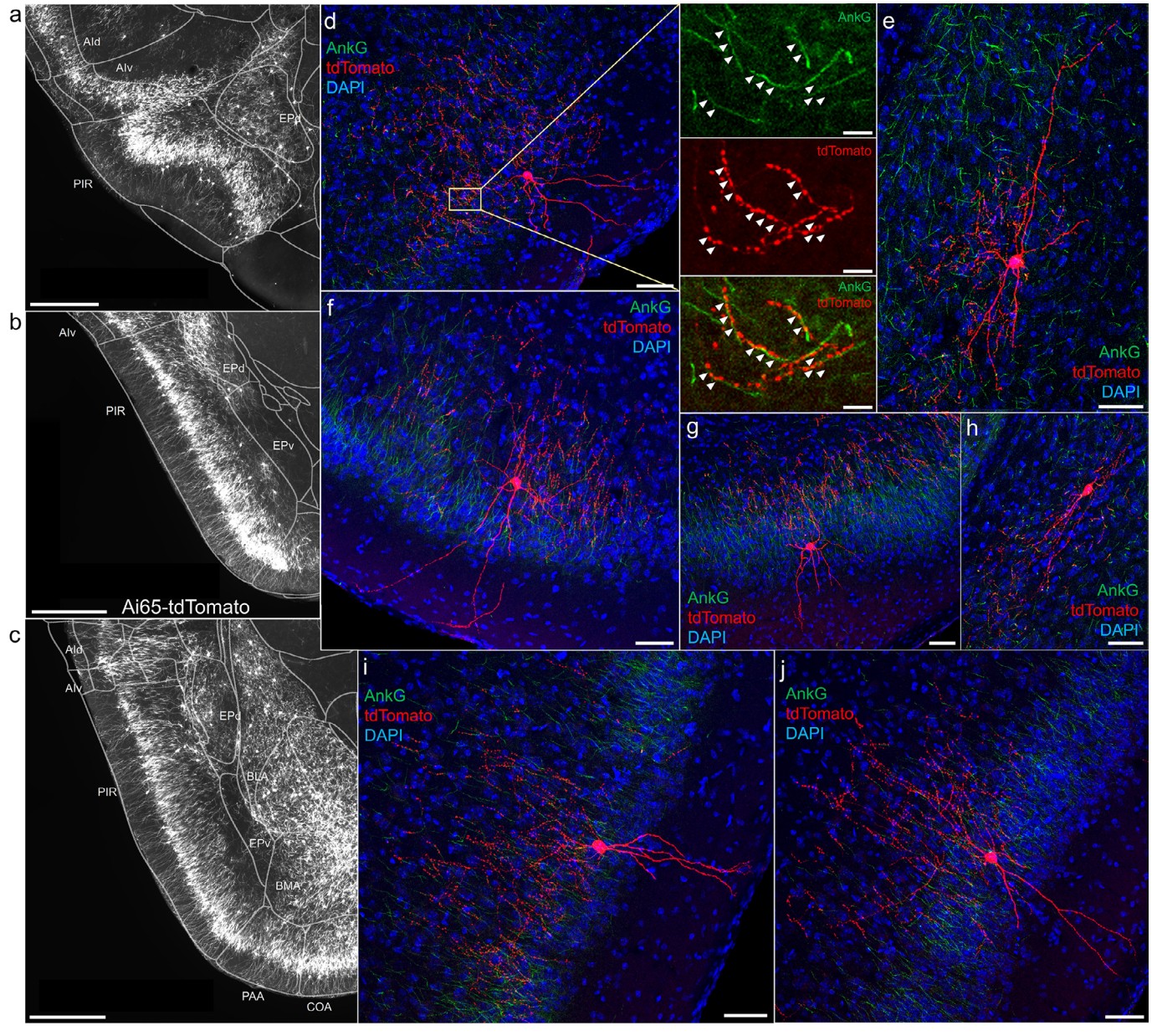

**Figure 5.** Axo-axonic cells (AACs) in piriform cortex. (**a–c**) Dense AAC labeling in piriform cortex, organized from anterior (**a**) to posterior (**c**) and overlaid with ARAv3 area outlines. Note the highly laminar distribution patterns. (**d-j**). Examples of single piriform AACs co-stained with AnkG (green). Note the relatively similar morphologies with only a few apical dendrites extending to the pial surface (**d-g, i-j**). An example of a deeper layer AAC at the border of piriform and EPv (**h**). White arrowheads in high-magnification insets indicate segments of overlap for RFP-labeled AAC and AnkG-labeled axon initial segment (AIS). Scale bars, 500 μm for grayscale images, 50 μm for confocal, 10 μm for high-magnification insets.

dendrites often extending to the pia and basally oriented axon arbors; their somata are located at the upper border of the pyramidal cell layer in the case of TT and AON (*Figure 6c–g*). Overall, AACs in piriform, TT, and AON exhibited higher density than AACs in cortex, hippocampus, and amygdala (*Figure 6h*, see Extended Data for *Figures 2, 3, and 6*).

## AACs in the lateral septum

The LS, despite having no distinct laminar structure (delineated in ARAv3), shows some evidence of differential mRNA expression in superficial and deep domains, possibly analogous to a laminar organization (*Besnard and Leroy, 2022*). In *Unc5b*^CreER; *Nkx2.1*^Flp; Ai65 mice, AACs tended to locate toward

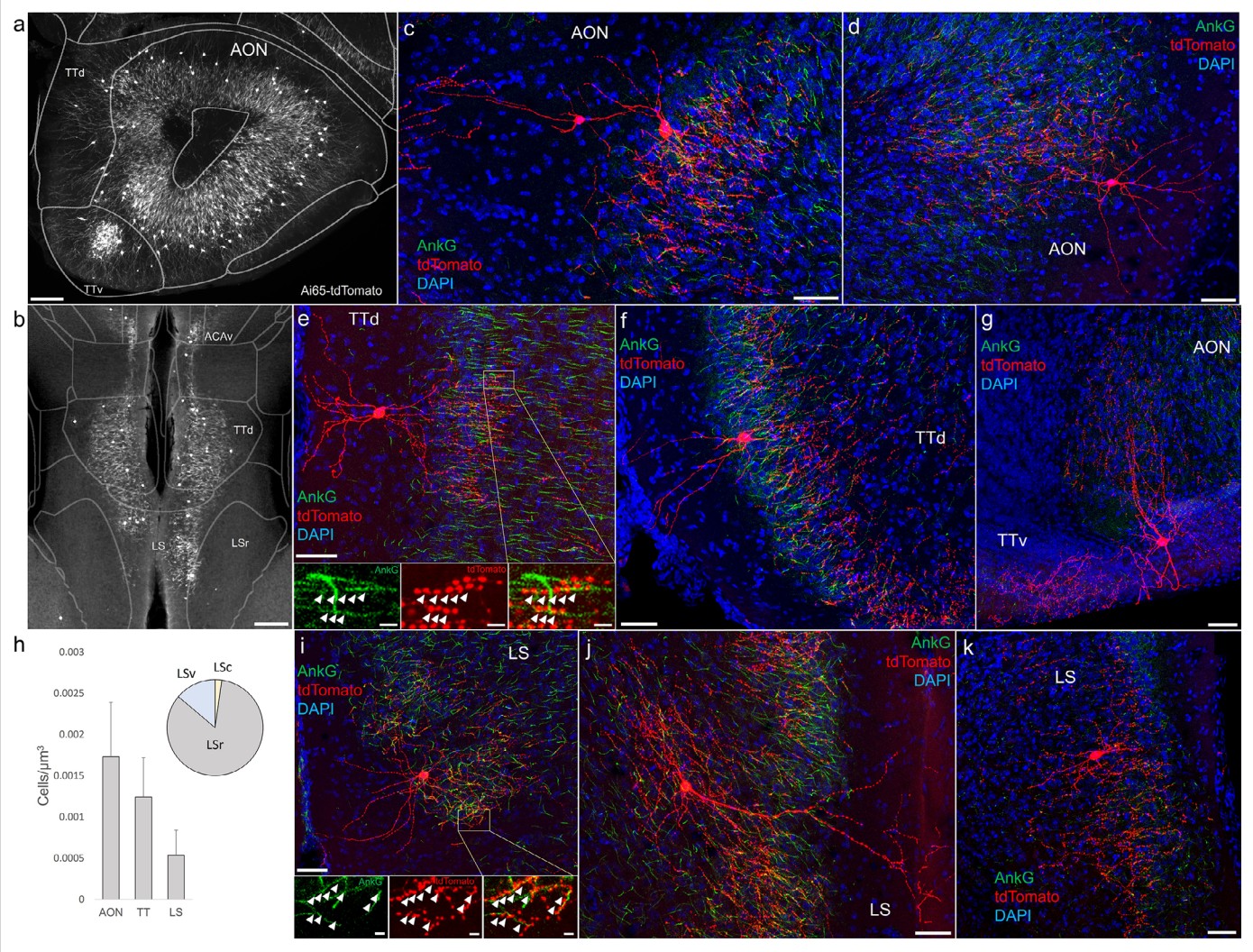

**Figure 6.** Identification and distribution of novel axo-axonic cell (AAC) subpopulations in anterior olfactory nucleus (AON), taenia tecta (TT), and septum. Dense population labeling in anterior AON and TT (**a**) and more posterior sections containing the TT and LS (**b**). Anteriorly, AACs are more concentrated in ventral TT while posteriorly in dorsal TT. Within LS, AACs tend to occupy the medial most portions along the midline. (**c–g, i–k**) Single-cell (confocal) labeling of AACs in AON (**c, d**), TT (**e–g**), and lateral septum (**i–k**). Note that despite areal variations, individual AACs preserve their stereotyped morphological characteristics (apical-oriented dendrite and basal-oriented axon arbor) that conform to laminar patterns of targets in each anatomical area. High-magnification insets: white arrowheads indicate segments of AnkG and RFP apposition. Scale bars, 200 µm for grayscale panels, 50 and 5 µm for single-cell examples and insets. (**h**) Normalized cell density (cells/µm³) for AACs in AON, TT, and lateral septum. Within lateral septum, the majority of AACs are located in rostral compartments (LSr). Data are mean ± standard error of the mean (SEM).

The online version of this article includes the following source data for figure 6:

**Source data 1.** Mean, SEM and percentages for data points in *Figure 6h*.

the midline (*Figure 6b*). Individual AACs were morphologically similar to cortical AACs, with extensive apical dendrites extending toward the midline and axon arbors and cartridges typically protruding away and into the cellular layer (*Figure 6i–k*).

## Monosynaptic input tracing to AACs in sensorimotor cortex

In addition to using Cre-AND-Flp-dependent reporter lines for AAC access, it is advantageous to extend our intersectional strategy to engage viral vectors that enable more selective and anatomy-based labeling, recording, and manipulation. Because the endogenous expression of *Unc5b* and *Nkx2.1* occur at non-overlapping stages in development, it is not feasible to directly use Cre-AND-Flp-dependent viral vectors (*Fenno et al., 2020*). Instead, we crossed *Unc5b*^CreER^; *Nkx2.1*^Flp^ driver with

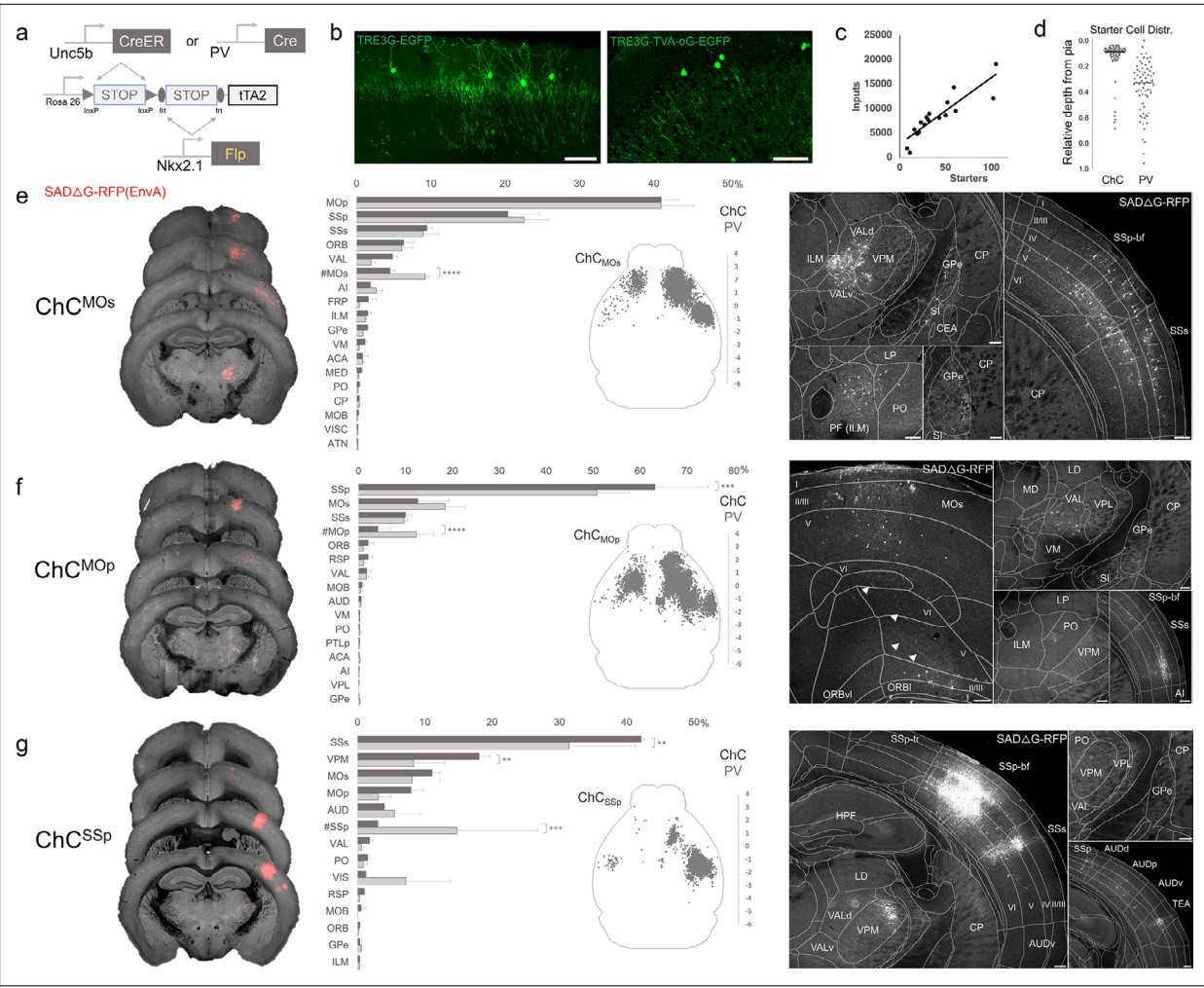

**Figure 7.** Intersectional tTA2 conversion enables viral access and input tracing to chandelier cells (ChCs) in sensorimotor cortices. (**a**) Schematic of intersectional tTA2-conversion strategy. Expression of *Nkx2.1*Flp and subsequent *Unc5b*CreER or *Pvalb*Cre results in activation of tTA2 expression in axo-axonic cells (AACs). This provides a genetic handle for versatile viral targeting and manipulation. (**b**) Cortical injection of TRE3G-promoter AAV vectors enables specific targeting of ChCs (green, left). A similar backbone was used to construct a TRE3G rabies starter AAV for monosynaptic input tracing, which contains both TVA and optimized rabies glycoprotein (right). Compared to TRE3G-EGFP, the TRE3G starter AAV has weaker EGFP labeling of neurites. Scale bar, 200 μm. (**c**) Linear regression fit of inputs cells/starter cells. Each point represents one animal, *n* = 18, *r* = 0.8874, slope = 135.51. (**d**) Relative pia-to-white matter depth distribution of starter cells (expressing both AAV starter and rabies). ChC starter cell tended to cluster closer to the pia while *Pvalb* cells were more broadly distributed throughout the cortical upper layers, reflecting the differences in the distribution of the two subtypes. (**e–g**) Synaptic input source to ChCs in primary motor area (MOp), secondary motor area (MOs) and primary somatosensory area (SSp), with *Pvalb* cells as comparison. Only long-range inputs (i.e. outside the ARAv3 area injected) are plotted. Left: Representative coronal sections from three indicated cortical areas overlaid with macroscale view of rabies tract labeling (in red, gray is autofluorescence) to each subpopulation of ChC. Middle: Histogram comparing percentage of inputs from each structure innervating the indicated cortical subpopulation of ChCs (dark gray) and *Pvalb* (light gray), sorted from largest input source to smallest. Inset: Representative top–down map of all inputs to AACs from a single animal. Right: Representative sections highlighting select sources of rabies-labeled inputs to each ChC population, overlaid with ARAv3 boundaries. Scale bars, 200 μm. For e–g, data are mean ± standard error of the mean (SEM), \*\*p < 0.01, \*\*\*p < 0.001, \*\*\*\*p < 0.0001 indicates p ≥ 0.05 (two-way analysis of variance [ANOVA] with post hoc Bonferroni correction); for scatterplot in d, median is plotted. Homotypic projections to injection site from contralateral hemisphere are denoted by (#). All images showing EGFP were immunostained for signal amplification, while rabies-RFP shown in e–g were native fluorescence. For each MOs, MOp, and SSp, *N* = 6 mice for (3 *Unc5b; Nkx2.1*, 3 *Pvalb; Nkx2.1*; total *N* = 18 mice).

The online version of this article includes the following source data and figure supplement(s) for figure 7:

**Source data 1.** Number of input cells, depth, mean, SEM and associated statistics for data points in *Figure 7c-g*.

**Figure supplement 1.** Intersectional tTA2 conversion enables viral access and input tracing to chandelier cells (ChCs) and *Pvalb* cells.

a Cre-AND-Flp-dependent tTA2 reporter line (*Figure 7a*; *Matho et al., 2021*), which provides an indelible genetic handle for delivery of tTA-activated viral reporters. We tested this approach in AACs to map their synaptic input source by using tTA-activated AAVs in of *Unc5b*[CreER]; *Nkx2.1*[Flp]; dual-tTA2 mice (*Figure 7b*; *Figure 7—figure supplement 1b, c*). We used the *Pvalb* cell (PVC) population for comparison largely due to convenience, recognizing the fact that PVCs also include some ChCs.

We crossed the dual-tTA2 reporter to either *Unc5b*[CreER]; *Nkx2.1*[Flp] or *Pvalb*[Cre]; *Nkx2.1*[Flp] and performed monosynaptic input tracing to ChCs and *Pvalb* cells (PVCs), respectively, in three sensorimotor areas: MOs, MOp, and SSp (*Figure 7a–c*, *Figure 7—figure supplement 1a*). Intersecting *Pvalb*[Cre] with *Nkx2.1*[Flp] is a strategy for specifically targeting *Pvalb* INs, as *Pvalb*[Cre] also captures a sparse population of excitatory pyramidal neurons (*van brederode et al., 1991*). Although *Pvalb* is expressed in up to ~50% of cortical ChCs, starter cell depth distribution was found to be largely non-overlapping between injected ChCs and PVCs, likely due to the much higher abundance of *Pvalb* basket cells (*Figure 7d*, *Figure 7—figure supplement 1d, e*).

Using STP tomography and our registration/cell detection pipeline, we mapped long-range inputs to different cortical ChC populations. Across sensorimotor cortex ChCs and PVCs integrated into highly interconnected cortical networks. ChCs[MOs] received ~50% of inputs from motor areas (41.0 ± 2.3% in MOp, 4.5 ± 0.7% in contra MOs) followed by SSp (20.4 ± 4.2%) and SSs (9.5 ± 0.7%) (*Figure 7e*). Long-range inputs to ChCs[MOp] were largely dominated by sensory cortex (63.2 ± 11.3% in SSp, 10.1 ± 1.3% in SSs), followed by MOs (12.7 ± 6.7%) and contra MOp (4.2 ± 2.5%) (*Figure 7f*). In contrast, ChCs[SSp] received only ~65% inputs total from sensorimotor (42.2 ± 0.5% in SSs, 11.1 ± 1.2% in MOs, 7.8 ± 1.9% in MOp, 3.03 ± 0.2% in contra SSp), with considerably more inputs from thalamus and other cortical areas (*Figure 7g*). Thalamocortical (TC) afferents to ChCs[MOs] and ChCs[MOp] were primarily from VAL, whereas TC inputs to ChCs[SSp] were mostly from ventral posteromedial nucleus of the thalamus (VPM). In addition to sensorimotor, other cortical areas such as ORB, FRP, AI, VIS, ACA, RSP, and AUD comprised a minor source of long-range inputs. Interestingly, we identified sparse inputs from GPe directly to ChCs and *Pvalb* in all three cortical areas. Overall input patterns were comparable between ChCs and *Pvalb*, although ChCs received significantly less homotypic inputs from the contralateral hemisphere for each injection site (*Figure 7e–g*).

## Monosynaptic input tracing to AACs in CA1

The functional role of AACs is perhaps best understood in hippocampus, where CA3 AACs tend to fire at distinct epochs compared to other cell types and near the peak of theta oscillations (*Viney et al., 2013*). Little is known about the inputs to AACs that might drive this cell-type functional distinction, although non-specific tracers have identified an inhibitory connection from medial septum (MS) to AACs in CA3 (*Viney et al., 2013*). More recently, we used *Unc5b*[CreER] to functionally characterize the firing patterns of AACs[CA1] during whisking and locomotion (*Dudok et al., 2021*). Here, we applied cell type targeted retrograde monosynaptic tracing to map long-range inputs to AACs[CA1] (*Figure 8a, b*).

Compared to cortical ChCs, AACs[CA1] received inputs from more diverse sets of sources. Within the hippocampus formation, AACs[CA1] received asymmetric inputs from each hemisphere, with relatively a higher fraction in CA3 contralaterally (29.0 ± 2.4%) but higher ipsilaterally in DG (4.3 ± 1.5%) and ENT (11.6 ± 3.6%) and SUB (15.2 ± 5.2%) (*Figure 8c*). Surprisingly, while the aforementioned study *Viney et al., 2013* found inhibitory connections from MS to AACs in CA3, we found in AACs[CA1] over twofold higher input connectivity from adjacent NDB (5.1 ± 0.9%) compared to MS (2.0 ± 0.2%) (*Figure 8d, e*). Immunohistochemistry analysis found the majority of these long-range inputs were inhibitory (56.8 ± 4.8%) and a sizeable proportion ChAT-positive (27.6 ± 4.0%) (*Figure 8f–j*). AACs[CA1] also received diverse cortical inputs, including from VIS, RSP, AUD, SSp, PTLp, SSs, Tea, and PERI, as well as TC afferents from ATN (*Figure 8l–p*; see abbreviations in *Supplementary file 1*).

## Monosynaptic input tracing to AACs in BLA

Comparatively little is known about the function or circuit role of AACs[BLA]. Past studies have had to largely rely on post hoc identification from paired recordings, demonstrating their potent inhibitory effects (*Veres et al., 2014*). However, without cell-type-specific tools for AACs[BLA], the long-range afferent connectivity that might drive these effects has remained virtually unknown. We applied our intersectional targeted approach to reveal the monosynaptic inputs to AACs[BLA] (*Figure 9a, b*). AACs[BLA] received highly diverse sets of inputs (*Figure 9c*). AACs[BLA] received strongest afferents from

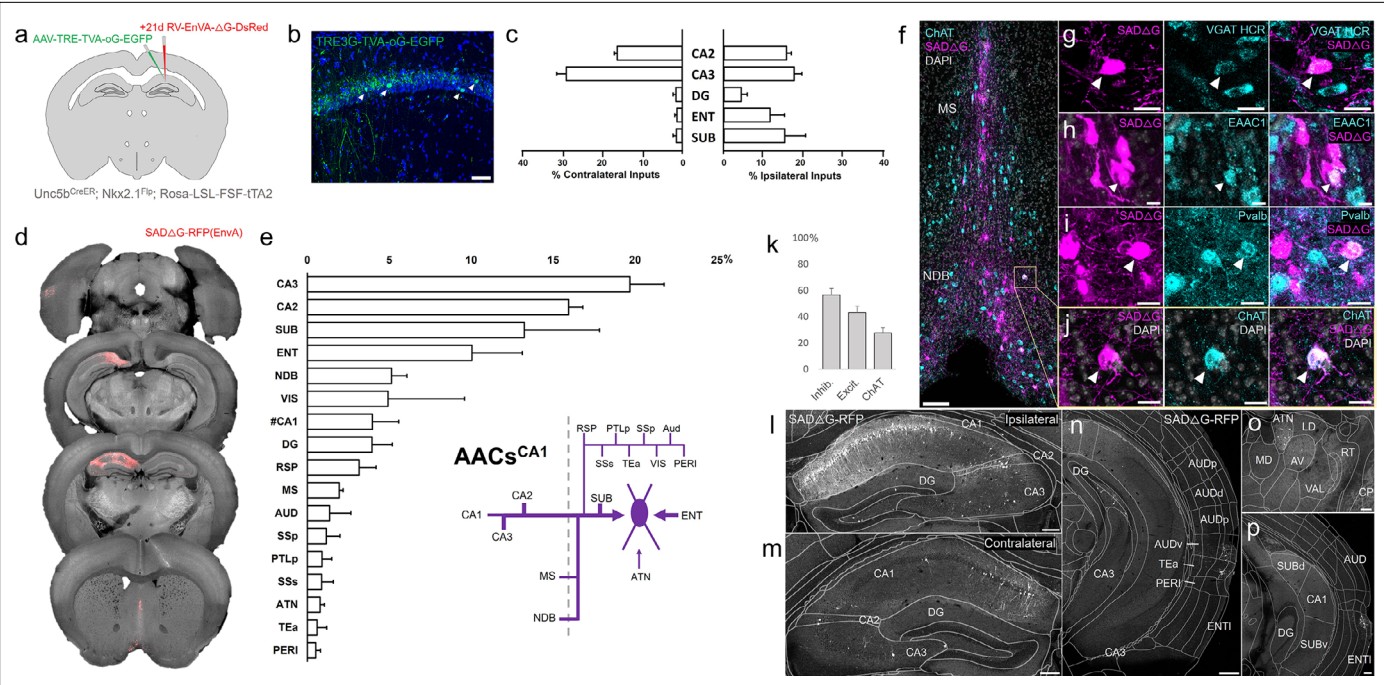

**Figure 8.** Monosynaptic input tracing to AACs$^{CA1}$ of hippocampus. (**a**) Schematic of retrograde synaptic tracing from CA1 axo-axonic cells (AACs) with site of viral injection. (**b**) Starter AAV expression in CA1 (green). White arrowheads show soma position is largely confined to the pyramidal cell layer. Scale bar, 50 μm. (**c**) Comparative input distribution from ipsilateral (right) and contralateral (left) hippocampal compartments. (**d**) Mesoscale rabies labeling of presynaptic input cells (red) to AACs$^{CA1}$ (**e**) Histogram of percentage of total inputs from each structure, sorted from largest to smallest. Inset: Summary schematic of input sources to AACs$^{CA1}$. AACs$^{CA1}$ receive strong innervation from entorhinal, subiculum, and contralateral hippocampus, with more minor innervation from thalamus, medial septum (MS), diagonal band nucleus, and various cortical areas. (**f–j**) Immunohistochemistry and hybridization chain reaction-fluorescence in situ hybridization (HCR RNA-FISH) marker analysis of input cells in MS and nucleus of the diagonal band (NDB), indicated by color-coded labels. Examples of co-labeled cells are shown following HCR RNA-FISH for the inhibitory marker vGAT (**g**) and immunohistochemistry for the excitatory marker EAAC1 (**h**), *Pvalb* (**i**) and ChAT (**j**). White arrowheads indicate co-labeled cells for each marker. Scale bars, 50 and 20 μm for high-magnification insets. (**k**) Quantification of marker colocalization with rabies labeling in MS and NDB. (**l–p**) Representative coronal sections highlighting select sources of rabies-labeled inputs to AACs$^{CA1}$, overlaid with ARAv3 atlas boundaries. Inputs within hippocampus were mostly from within CA1 both ipsilateral (**l**) and contralateral (**m**) to the injection site. Input cells were also found in adjacent cortical areas more posterior such as entorhinal cortex (ENT), auditory cortex (AUD), and temporal association area (TEa) (**n–p**). Thalamic inputs were largely confined to the anterior group of the dorsal thalamus (ATN) (**o**). Scale bars, 200 μm. For c, e, and k, data are mean ± standard error of the mean (SEM) from *N* = 4 mice. Homotypic projections to injection site from contralateral hemisphere are denoted by (#CA1) in e. EGFP in b and rabies-RFP in f–j were immunostained for signal amplification, while rabies-RFP shown in l–p were native fluorescence.

The online version of this article includes the following source data for figure 8:

**Source data 1.** Mean and SEM for data points in (*Figure 8c, e, k*).

PIR (13.1 ± 1.1%) and the hippocampal–entorhinal network (ENT 12.7 ± 0.2%, CA1 10.5 ± 2.0%, SUB 8.0 ± 0.1%) (*Figure 9e, h–i*). The next strongest inputs originated from cortical amygdala areas (TR, PAA, COA) and cortical subplate nuclei (CeA, LA, EP, BMA) (*Figure 9d*). TC afferents were largely from MTN (2.6 ± 0.4%), while cortical projections were identified in AI, AUD, PERI, and ECT (*Figure 9f*). Input cells in LZ and MEZ of hypothalamus were found to be glutamatergic, while those from GPe and EP were GABAergic (*Figure 9j–o*).

## Discussion

### Specific and comprehensive targeting of ground truth cell types

A major challenge in studying the functional organization of GABAergic inhibitory circuits, and neural circuits in general, is establishing specific and comprehensive genetic access to ground truth cell types. Here, leveraging an intersectional strategy that integrates lineage and molecular markers, we have achieved near-perfect specificity and near-complete comprehensiveness in targeting a bona fide IN type. This strategy will facilitate studying the connectivity, function, and development of AACs

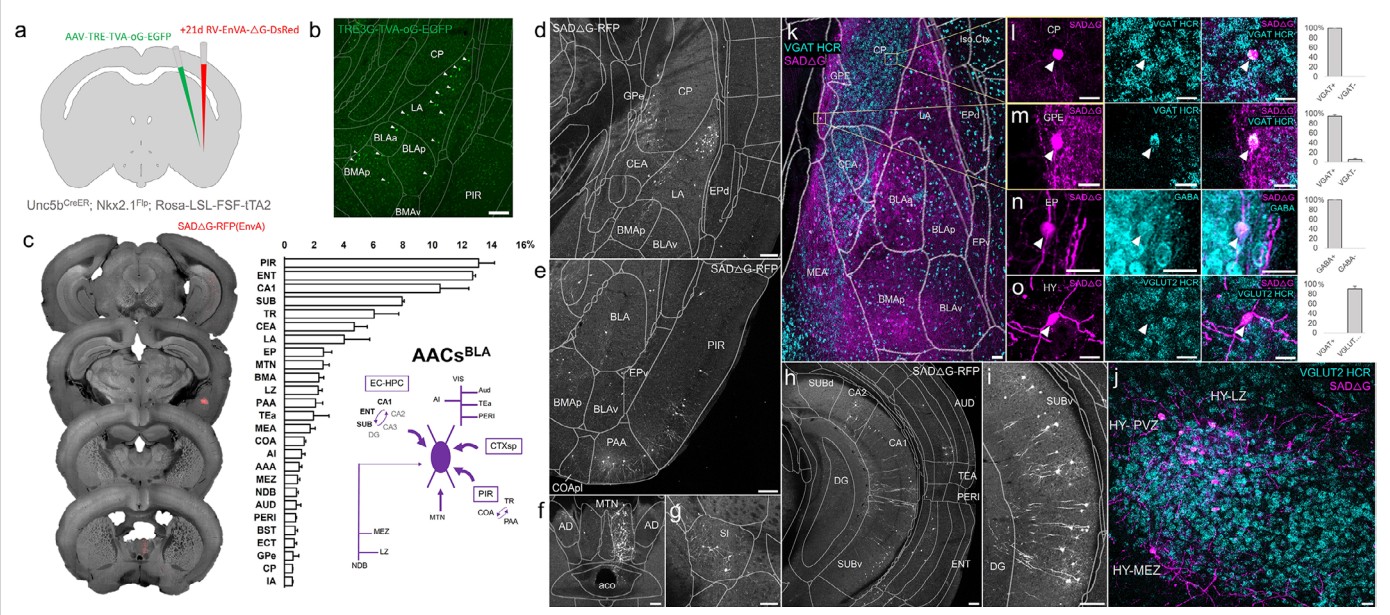

**Figure 9.** Monosynaptic input tracing to AACs^BLA. (**a**) Schematic of retrograde synaptic tracing from basolateral amygdala (BLA) axo-axonic cells (AACs) with site of viral injection. (**b**) Starter AAV distribution as indicated by white arrowheads, overlaid with ARAv3 atlas. Scale bar, 200 μm. (**c**) Left: Mesoscale rabies labeling of presynaptic input cells (red) to AACs^BLA. Right: Histogram of percentage of total inputs from each structure, sorted form largest to smallest. Inset: Summary schematic synaptic input source to AACs^BLA. AACs^BLA are innervated by diverse sets of inputs, with strongest innervation from entorhinal cortical–hippocampal network (EC-HPC), piriform areas (PIR), cortical subplate (CTXsp), and minor subcortical and cortical inputs. (**d–j**) Representative coronal sections of rabies-labeled inputs (gray) across input structures, overlaid with ARAv3 boundaries. Scale bars, 200 μm. Inputs to AACs^BLA were found in amygdalar and cortical-amygdalar areas (**d, e**), piriform (**e**), midline group of the dorsal thalamus (**f**), substantia innominata (**g**), posterior hippocampus and cortex (**h**), and the ventral subiculum (**i**). Examples of dense input cells in medial (MEZ) and lateral (LZ) hypothalamus overlaid with excitatory vGLUT2 HCR RNA-FISH (**j**). (**k–o**) Immunohistochemistry and RNA-FISH marker analysis of input cells in striatum/CP, GPe, EP, and hypothalamus. VGAT FISH or GABA IHC were used for identifying GABAergic input cells (**l–n**) while VGLUT2 FISH was used an excitatory marker (**o**) as indicated. White arrowheads indicate co-labeled cells. Scale bars, 50 and 20 μm for high-magnification insets. Right: Quantification of marker colocalization with rabies labeling for each corresponding area. Inputs to AACs^BLA from CP, GPe, and EP were inhibitory, while inputs from hypothalamus were excitatory. Data are mean ± standard error of the mean (SEM) from N = 3 mice. Rabies-RFP in j–o were immunostained for signal amplification, while EGFP in b and rabies-RFP shown in d–i were native fluorescence.

The online version of this article includes the following source data for figure 9:

**Source data 1.** Mean and SEM for data points in *Figure 9c, l-o*.

---

across brain regions, thereby providing a robust experimental platform for integrating multi-modal analysis of a ground truth cell type across brain circuits. Both *Pthlh*^Flp- and *Unc5b*^CreER-mediated intersectional approaches are also useful for studying the development of AACs. We recognize that it is currently not possible to fully assess the 'completeness' of AAC targeting; this may require a comparison with complete labeling by anatomical ground truth, such as dense EM reconstruction of all AACs across the brain volume. Another practical drawback of our approach is the requirement of combining three engineered genetic alleles. In this context, the *Unc5b*^CreER line by itself is already useful for AAV-mediated targeting of AACs (*Dudok et al., 2021*). Because most AAVs are neurotrophic, Cre-activated AAVs would bypass endothelial cells to target only AACs in the *Unc5b*^CreER line, even though this approach is not suited to quantify brain-wide AAC patterns. Beyond relying on germline engineering, the recently invented CellREADR technology (*Qian et al., 2022*) will further simplify AAC targeting by identifying an RNA sensor of *Unc5b* mRNA and deliver it with an AAV vector. As *Unc5b* is a conserved AAC marker from rodents to primates including humans, the CellREADR approach may enable genetic access to AACs across mammalian species.

## Implications of brain-wide AAC distribution pattern

The initial discovery of neocortical AACs was based on their striking chandelier-like morphology (*Szentágothai, 1975*), reflecting the spatial arrangement of their exclusive synaptic targets – the AIS

of cortical pyramidal neurons (*Somogyi, 1977*; *Somogyi et al., 1982*). This defining feature of synapse specificity led to subsequent discoveries of AACs in the hippocampus and BLA. These findings have relied on meticulous efforts of skilled neuroanatomists to correctly identify AACs, mostly post hoc, by synapse specificity at AIS (*Bienvenu et al., 2012*; *Veres et al., 2014*). Since then, it has been widely assumed that AACs are restricted to these brain structures, and likely to the mammalian species. Here, based on developmental lineage and key molecular markers that likely include all AACs, our results reveal a vastly expanded and likely complete brain distribution pattern of AACs. Most importantly, we found that pAACs are deployed to the entire pallium-derived brain structures (*Puelles, 2017*), including not only the dPAL-derived neocortex and mPAL-derived hippocampal formation, but also the lPAL-derived claustrum–insular complex, the vPAL-derived extended amygdaloid complex, piriform cortex, and other olfactory centers such as AON and TT. In addition, subregions of the lateral septum and hypothalamus also contain pAACs. In this context, it is notable that the pallium may have its evolutionary origin in an anterior region of the vertebrate hypothalamus (*Puelles and Rubenstein, 2003*).

Among the pallium-derived structures, the density of AACs vary substantially within and across regions. For example, within the neocortex, ChCs are highest in perirhinal and lowest in retrosplenial cortex. Across cortical layers, AACs are highly concentrated in L2, enriched in L5/6, and significantly less abundant in middle layers. Consistent with these results, the number and density of GABAergic synapses at AIS also vary substantially among different PyNs within and across cortical layers (*DeFelipe et al., 1985*; *Fish et al., 2013*; *Schneider-Mizell et al., 2021*). Similar variations are seen across the hippocampal formation, where CA2 exhibit the highest and DG the lowest density. It is also possible that these regional differences might reflect the distinction of PN types and/or their inherent physiological and connectivity properties that might require different levels and ratio of AAC control.

In particular, the finding of a striking abundance of AACs in olfactory centers across piriform cortex, AON, and TT suggests strong inhibitory regulation of PNs at AIS in these regions. Although the functional significance is not yet clear, the simpler circuit architecture and strong link to innate rodent behaviors of these olfactory centers may facilitate uncovering the logic of AAC function. Furthermore, as these olfactory circuits are evolutionarily preserved across vertebrate species (*Zeppilli et al., 2023*), our finding raises the possibility that AACs might be present in the CNS of other vertebrates, beyond the mammalian brain.

Notably, although the MeA and BST comprise mostly GABAergic neurons that are primarily generated from the subpallium, they contain subsets of glutamatergic PN that migrate from pallium origin (*Beyeler and Dabrowska, 2020*) or from the third ventricle (*García-Moreno et al., 2010*). The true AAC identity in these regions, and in several other regions such as the hypothalamus, remains to be validated by synapse innervation at AIS of GLU PNs. Nevertheless, based on current results, our overarching hypothesis is that AACs are deployed across all pallium-derived structures to control the AIS of GLU PNs, including those in MeA and BST. Our results also indicate that AACs are absent from the rest of the CNS. This finding suggests a major distinction between the physiology and functional organization of pallium-derived GLU PNs versus those in other CNS regions.

## AACs as a model for understanding neuronal cell-type organization across regions and species

The rapid scaling of single-cell RNA sequencing enables the generation of massive transcriptome datasets across brain regions and species (*Bakken et al., 2021*; *Hain et al., 2022*). These datasets provide unprecedented opportunities to discover the organization and relationship of cell types within and across brain regions, which reflect their gene regulatory programs and developmental origin. Cross species analyses promise to further reveal the evolutionary conservation and divergence of brain cell types (*Bakken et al., 2021*; *Tosches, 2021*). However, transcriptomic cell homologies between brain regions and species, especially more distantly related regions and species, are often discernible only at a relatively coarse level and do not fully capture the biological complexity (*Tosches, 2021*). A major challenge is to evaluate statistically defined transcriptomic similarity against biological ground truth, such as anatomic connectivity and developmental relationship. In this context, the most appealing attribute of AACs is their singularly defining synapse specificity at AIS of GLU PNs, which can be readily measured and validated. Under this overarching connectivity-based definition, all other molecular and phenotypic variations can be assessed and organized accordingly regardless of brain

regions and species. Therefore, genetic access to AACs across brain regions provides a rare and compelling ground truth platform that can unambiguously identify this cell type across regions and species. This will further facilitate transcriptomic and epigenomic analysis to discover the gene regulatory program of AACs and contribute to understanding the principles of cell-type organization by linking developmental genetic program, circuit connectivity and function, and evolutionary trajectory.

# Materials and methods

## Key resources table

| Reagent type (species) or resource | Designation | Source or reference | Identifiers | Additional information |
|---|---|---|---|---|
| Antibody | Anti-AnkryinG (mouse monoclonal) | Neuromab | Cat#: 75-146 | 1:500 |
| Antibody | Anti-ChAT (goat polyclonal) | Millipore | Cat#: AB144P | 1:300 |
| Antibody | Anti-EAAC1 (goat polyclonal) | Millipore | Cat#: AB1520 | 1:500 |
| Antibody | Anti-GABA (rabbit polyclonal) | Sigma | Cat#: A2052 | 1:500 |
| Antibody | Anti-GFP (chicken polyclonal) | Aves Labs | Cat#: 1020 | 1:1000 |
| Antibody | Anti-parvalbumin (mouse monoclonal) | Sigma | Cat#: P3088 | 1:1000 |
| Antibody | Anti-Phospho-I$\kappa$B$\alpha$ (rabbit polyclonal) | Cell Signaling Technology | Cat#: 2859 | 1:500 |
| Antibody | Anti-RFP (rabbit polyclonal) | Rockland | Cat#: 600-401-379 | 1:1000 |
| Commercial assay or kit | HCR v3.0 | Molecular Instruments | | |
| Other | AAVDJ-TRE3g-EGFP | Vigene | | Adeno-associated virus, titer: $5 \times 10^{12}$ genome copies (gc)/ml |
| Other | AAVDJ-TRE3g-TVA-oG-EGFP | Vigene | | Adeno-associated virus, titer: $9.24 \times 10^{12}$ genome copies (gc)/ml |
| Other | G-Deleted Rabies dsRedXpress | Salk Institute | | Rabies virus, titer: $5.0 \times 10^{7}$ colony-forming units (cfu)/ml |
| Software, algorithm | ImageJ | NIH | RRID:SCR_003070 | |
| Software, algorithm | MATLAB | Mathworks | RRID:SCR_001622 | R2017b |
| Software, algorithm | Elastix | Image Sciences Institute | RRID:SCR_009619 | |
| Strain (*Mus musculus*) | Ai65 | The Jackson Laboratory | RRID:IMSR_JAX:021875 | |
| Strain (*M. musculus*) | C57BL/6J | The Jackson Laboratory | RRID:IMSR_JAX:000664 | |
| Strain (*M. musculus*) | dual-SynaptotagminEGFP-2A-tdTomato | The Jackson Laboratory | RRID:IMSR_JAX:030206 | |
| Strain (*M. musculus*) | dual-tTA2 | The Jackson Laboratory | RRID:IMSR_JAX:036304 | |
| Strain (*M. musculus*) | Nkx2.1$^{Cre}$ | The Jackson Laboratory | RRID:IMSR_JAX:008661 | |
| Strain (*M. musculus*) | Nkx2.1$^{FlpO}$ | The Jackson Laboratory | RRID:IMSR_JAX:028577 | |
| Strain (*M. musculus*) | Pthlh$^{FlpO}$ | This paper | | |
| Strain (*M. musculus*) | Pvalb$^{Cre}$ | The Jackson Laboratory | RRID:IMSR_JAX:017320 | |
| Strain (*M. musculus*) | Unc5b$^{CreER}$ | *Dudok et al., 2021* | | |

## Animals

All experiments were carried out in accordance with the guidelines of the Animal Care and Use Committee of Cold Spring Harbor Laboratory (CSHL). Mice were housed in groups of five per cage or singly if pregnant under standard vivarium conditions with a 12-hr light/dark cycle. *Unc5b*$^{CreER}$ (*Dudok et al., 2021*) or *Pvalb*$^{Cre}$ (JAX, Stock 017320) were crossed to *Nkx2.1*$^{Flp}$ (JAX, Stock 028577). *Pthlh*$^{Flp}$ (unpublished) was crossed to *Nkx2.1*$^{Cre}$ (JAX, Stock 008661). For each Cre/Flp combination, offspring hemizygous for both alleles were crossed to Ai65 (JAX, Stock 021875), Rosa26-loxpSTOPloxp-frtSTOPfrt-synaptotagminEGFP-2A-tdTomato (JAX, Stock 030206), or Rosa26-loxpSTOPloxp-frtSTOPfrt-tTA2 (JAX, Stock 036304) reporter mice. Mice with *Unc5b*$^{CreER}$ were induced by tamoxifen i.p. (TM; Sigma-Aldrich, Cat# T5648) administered either (1) for dense labeling and input tracing, 100 mg/kg for 5 consecutive days starting at P55–60 (2) for single-cell labeling, 25 mg/kg for 1–2 consecutive days at P55–P60. The dosage and regimen of tamoxifen for dense AAC labeling were the maximum based on established protocols for conditional knockouts and one that was well-tolerated.

For single-cell labeling, the tamoxifen dosage was determined empirically for reliable labeling of non-overlapping AACs in different brain areas. Animals were fed ad libitum and their health status were routinely monitored. Both male and female animals were used in all experiments, with three to four mice total for each injection site × genotype and for cell distribution analysis. Sample sizes were estimated on the basis of previous studies using similar methods and analyses (*Oh et al., 2014*, *Kim et al., 2015*, *Harris et al., 2019*, *Matho et al., 2021*). Genotyping was performed using PCR and according to supplier protocols.

## Immunohistochemistry

Mice were anesthetized with Avertin and transcardially perfused with 0.9% saline followed by 4% paraformaldehyde in 0.1 M phosphate buffer. Following overnight post-fixation, brains were rinsed in phosphate-buffered saline (PBS) and sectioned at 50–65 µm thickness with a Leica VT1000S vibratome. For immunohistochemistry, sections were first treated with a secondary-matched blocking solution (10% normal goat or donkey serum and 0.2% Triton X-100 in PBS) for 1 hr, then incubated overnight at 4°C with primary antibodies diluted in 5% blocking solution. Sections were washed three times in PBS and incubated for 2 hr at room temperature with corresponding secondary antibodies, Goat or Donkey Alexa Fluor 488, 568, or 647 (1:500, Life Technologies) and 4',6-diamidino-2-phenylindole (DAPI) to label nuclei (1:1000 in PBS, Life Technologies, Cat#: 33342). Sections were washed three times with PBS and dry-mounted on slides using Fluoromount-G (SouthernBiotech, Cat#: 0100-01) mounting medium. Primary antibodies used were Anti-AnkryinG (1:500, Neuromab, Cat#: 75-146), Anti-ChAT (1:300, Millipore, Cat#: AB144P), Anti-EAAC1 (1:500, Millipore, Cat#: AB1520), Anti-GABA (1:500, Sigma-Aldrich, Cat#: A2052), Anti-GFP (1:1000, Aves Labs, Cat#: 1020), Anti-parvalbumin (1:1000, Sigma-Aldrich, Cat#: P3088), Anti-Phospho-IκBα (1:500, Cell Signaling Technology, Cat#: 2859), and Anti-RFP (1:1000, Rockland, Cat#: 600-401-379). Samples for AAC validation were collected from six mice given 25 mg/kg for 1–2 consecutive days, as described above. Validation was conducted with high-magnification confocal microscopy and defined by a cell exhibiting at least two RFP-labeled axons bearing multiple synaptic boutons aligned with AIS labeled by AnkryinG or Phospho-IκBα.

## Floating section two-color in situ hybridization

In situ hybridization was performed using hybridization chain reaction (HCR v3.0, Molecular Instruments). Samples post-fixed in 4% paraformaldehye (PFA) at 4°C were cryoprotected in 30% sucrose solution in RNAse-free diethyl pyrocarbonate-PBS (DEPC-PBS) at 4°C for 48 hr, frozen in Tissue-Tek O.C.T. Compound (Sakura), and stored at −80°C until sectioning. 50 µm thick coronal floating sections were collected into a sterile 24-well plate in DEPC-PBS, fixed again briefly for 5 min in 4% PFA, then placed in 70% EtOH in DEPC-PBS overnight. Sections were rinsed in DEPC-PBS, incubated for 45 min in 5% sodium dodecyl sulfate (SDS) in DEPC-PBS, rinsed and incubated in 2× sodium chloride-sodium citrate buffer (SSCT), pre-incubated in HCR hybridization buffer at 37°C, and then placed in HCR hybridization buffer containing RNA probes overnight at 37°C. The next day, sections were rinsed 4 × 15 min at 37°C in HCR probe wash buffer, rinsed with 2× SSCT, pre-incubated with HCR amplification buffer, then incubated in HCR amplification buffer containing HCR amplifiers at room temperature for ~24 hr. On the final day, sections were rinsed in 2× SSCT, rinsed again with 2× SSCT, then mounted on slides and coverslipped with Fluoromount-G (Southern Biotech).

## Microcopy and image analysis

Confocal imaging was performed on a Zeiss LSM 780 (CSHL St. Giles Advanced Microscopy Center) using ×10, ×20, or ×63 objectives. Imaging for grayscale overview images was performed on a Zeiss Axioimager M2 System equipped with MBF Neurolucida Software (MBF) using ×5, ×10, or ×20 objectives. Quantification and image analysis was performed using Image J/FIJI software with experimenter blind to genotype and experiment. Statistics and plotting of graphs were done using GraphPad Prism 7 and Microsoft Excel 2010. All images shown in figures were acquired by one-photon microscopy (confocal, epifluorescence) with the exception of *Figure 1—figure supplement 1g–n*, *Figure 7e–g*, *Figure 8l–p*, and *Figure 9d–i* which were acquired by STP tomography.

## Viruses

To construct AAVDJ-TRE3g-TVA-oG-eGFP, TVA, the optimized rabies glycoprotein oG (*Kim et al., 2016*), and eGFP were linked by 'self-cleaving' 2A and P2A peptides and cloned into pAAV-TRE3G (Sacha Nelson). For AAVDJ-TRE3g-eGFP only eGFP was cloned. AAVs serotype DJ were packaged by a commercial vector core facility (Vigene). Genomic titers of AAVDJ-TRE3g-TVA-oG-eGFP and AAVDJ-TRE3g-eGFP were $9.24 \times 10^{12}$ and $5.0 \times 10^{12}$ gc/ml, respectively. TVA, oG, and eGFP were subcloned from AAV-hSyn-FLEX-TVA-P2A-EGFP-2A-oG (Addgene, Cat# 85225). AAVs were diluted 1:5 in 1× PBS immediately prior to use. EnvA-pseudotyped G-Deleted Rabies dsRedXpress was purchased from Salk Institute ($5.0 \times 10^7$ titer units (TU)/ml).

## Stereotactic viral injections

Adult mice were anaesthetized with 2% isofluorane delivered by constant air flow (0.4 l/min). Analgesics ketoprofen (5 mg/kg) and dexamethasone (0.5 mg/kg) were administered subcutaneously just prior to sugery, while lidocaine (2–4 mg/kg) was applied intra-incisionally. Mice were mounted on a stereotaxic headframe (Kopf Instruments, 940 series or Leica Biosystems, Angle Two) and stereotactic coordinates identified as follows: (MOs): 2.25 mm anterior to bregma, 1.2 mm lateral, 0.3 mm in depth; (MOp): 0.5 mm anterior to bregma, 1.6 mm lateral, 0.3 mm in depth; (SSp): 1.12 mm posterior to bregma, 3.25 mm lateral, 0.3 mm in depth; (CA1): 1.94 mm posterior to bregma, 1.4 mm lateral, 1.4 mm in depth; (BLA): 1.9 mm posterior to bregma, 3.38 mm lateral, 4.76 mm in depth. An incision was made over the scalp, a small burr hole drilled in the skull and brain surface exposed. A pulled glass pipette with a tip of 20–30 µm containing virus was lowered into the brain. Virus was delivered at a rate of 30 nl/min using a Picospritzer (General Valve Corp) for a total volume of 100–150 nl for starter AAV and 300–400 nl for rabies virus. The pipette remained in place for 10 min prior to retraction to prevent backflow, after which the incision was closed with 5/0 nylon suture thread (Ethilon Nylon Suture Ethicon) or Tissueglue (3M Vetbond). Mice were kept warm on a heating pad until complete recovery. For input tracing experiments, *Unc5b*^CreER; *Nkx2.1*^Flp; dual-tTA2 (tamoxifen-induced as described above) and *Pvalb*^Cre; *Nkx2.1*^Flp; dual-tTA2 were injected at respective coordinates with AAVDJ-TRE3g-TVA-oG-eGFP at P75 ±2. After a 2- to 3-week incubation period, a second injection at the same coordinates was performed using EnvA-pseudotyped G-Deleted Rabies-dsRedXpress. Following another 7-day incubation mice were perfused as described above.

## Whole-brain STP tomography and image analysis

We used the whole-brain STP tomography pipeline described previously (*Ragan et al., 2012*; *Kim et al., 2015*; *Matho et al., 2021*). Perfused and post-fixed brains were embedded in 4% oxidized-agarose in 0.05 M phosphate buffer (PB), cross-linked in 0.2% sodium borohydrate solution (in 0.05 M sodium borate buffer, pH 9.0–9.5). The entire brain was imaged coronally with a 20×Olympus XLUMPLFLN20XW lens (NA 1.0) on a TissueCyte 1000 (Tissuevision) with a Chameleon Ultrafast-2 Ti:Sapphire laser (Coherent). EGFP/EYFP or tdTomato/dsRedXpress signals were excited at 910 or 920 nm, respectively. Whole-brain image sets were acquired as series of 12 (*x*) × 16 (*y*) tiles with 1 µm × 1 µm sampling for 230–270 z sections with a 50-µm z-step size. Images were collected by two PMTs (PMT, Hamamatsu, R3896), for signal and autofluorescent background, using a 560-nm dichroic mirror (Chroma, T560LPXR) and band-pass filters (Semrock FF01-680/SP-25). The image tiles were corrected to remove illumination artifacts along the edges and stitched as a grid sequence. Image processing was completed using ImageJ/FIJI and Adobe/Photoshop software with linear level and nonlinear curve adjustments applied only to entire images. Registration of brain-wide datasets to the Allen reference Common Coordinate Framework version 3 (CCFv3) was performed by 3D affine registration followed by a 3D B-spline registration using Elastix software (*Klein et al., 2010*), according to established parameters. Prior to STP analysis, datasets were screened for standard quality control according to pre-established criteria. Somata labeled by Ai65-tdtomato or rabies-dsRedXpress and catridges labeled by synaptotagmin-EGFP were automatically detected by a convolutional network trained as described previously (*Ragan et al., 2012*; *Kim et al., 2017*; *Matho et al., 2021*). For AAC laminar depth, somata and cartridge coordinates were overlaid on a mask for relative cortical depth, as described. For area cell density and cortical depth analysis, we registered each dataset, reporting cells detected in each brain structure without warping the imaging channel. For rabies input tracing experiments, only inputs originating outside the ARAv3 area injected were plotted (i.e. only

long-range inputs). All cell density quantifications are reported as mean cells per cubic μm ± standard error of the mean from $n = 4$ mice.

## Statistical analysis

Statistical analyses were done using GraphPad Prism (GraphPad Software Inc, San Diego, CA, USA). Normality of the data was assessed using Shapiro–Wilk and Smirnov–Kolmogorov tests. Detailed information for statistical tests performed, p-values, and sample sizes, and other descriptive statistics is included in the text (figure legends) and/or in the source data.

Parametric tests used: two-way analysis of variance with post hoc Bonferroni correction, Welch's *t*-test, Pearson's correlation coefficient. p-value criteria: *$p \leq 0.05$, **$p \leq 0.01$, ***$p \leq 0.001$, ****$p \leq 0.0001$.

## Acknowledgements

We thank P Wu for generating the *Unc5b*CreER and *Pthlh*Flp mouse lines, R Drewes for extensive support with STP imaging, K Matho for help with cell detection and image analysis, S George and L Van Aelst for helpful discussions and providing antibodies, and S Suryanarayana and D Huilgol for extensive discussion and critical reading of the manuscript. We thank CSHL Animal Resources for mouse husbandry and Salk for providing rabies virus. This work was supported in part by the NIH grants 5R01MH094705-10, 5R01MH109665-05, and 5DP1MH129954-02 to ZJH; 5R01MH101214-08 and 2R01MH108924-06 to B Li; and by the CSHL Robertson to ZJH. RR was supported by the NRSA F31 Predoctoral Fellowship 5F31MH114529-03 and AP by a NARSAD Post-doctoral Fellowship.

## Additional information

### Funding

| Funder | Grant reference number | Author |
|---|---|---|
| National Institute of Mental Health | 5R01MH094705-10 | Z Josh Huang |
| National Institute of Mental Health | 5R01MH109665-05 | Z Josh Huang |
| National Institute of Mental Health | 5DP1MH129954-02 | Z Josh Huang |
| National Institute of Mental Health | 5R01MH101214-08 | Bo Li |
| National Institute of Mental Health | 2R01MH108924-06 | Bo Li |
| CSHL Robertson | | Z Josh Huang |
| National Institute of Mental Health | 5F31MH114529-03 | Ricardo Raudales |
| National Alliance for Research on Schizophrenia and Depression | Post-doctoral Fellowship | Anirban Paul |

The funders had no role in study design, data collection, and interpretation, or the decision to submit the work for publication.

### Author contributions

Ricardo Raudales, Data curation, Formal analysis, Funding acquisition, Validation, Investigation, Visualization, Methodology, Writing – original draft, Writing – review and editing; Gukhan Kim, Joshua Hatfield, Data curation, Formal analysis, Investigation; Sean M Kelly, Anirban Paul, Data curation, Formal analysis, Investigation, Writing – review and editing; Wuqiang Guan, Data curation, Investigation; Shengli Zhao, Yongjun Qian, Data curation, Validation; Bo Li, Z Josh Huang, Conceptualization,

Resources, Supervision, Funding acquisition, Methodology, Project administration, Writing – review and editing

## Author ORCIDs

Ricardo Raudales http://orcid.org/0009-0006-2874-2972
Anirban Paul http://orcid.org/0000-0001-5347-9260
Bo Li https://orcid.org/0000-0002-0154-3088
Z Josh Huang https://orcid.org/0000-0003-0592-028X

## Ethics

All experiments were carried out in accordance with the guidelines of the Animal Care and Use Committee of Cold Spring Harbor Laboratory (CSHL). All of the animals were handled according to approved Institutional Animal Care and Use Committee (IACUC) protocols (#19-16-13-09-8) of Cold Spring Harbor Laboratory. All surgery was performed under isoflurane anesthesia and every effort was made to minimize suffering.

Reviewer #2 (Public Review): https://doi.org/10.7554/eLife.93481.3.sa1
Reviewer #3 (Public Review): https://doi.org/10.7554/eLife.93481.3.sa2
Author response https://doi.org/10.7554/eLife.93481.3.sa3

---

# Additional files

## Supplementary files

- MDAR checklist
- Supplementary file 1. List of Allen Reference Area v3 (ARAv3) brain area labels and abbreviations.

## Data availability

Source data for AAC and input tracing cell quantification in Figures 2—4, 6—9 are provided in a source data file. Registration of whole-brain images to the Allen reference Common Coordinate Framework version 3 (CCFv3) was performed by 3D affine registration followed by a 3D B-spline registration using Elastix software (Klein et al., 2010). Somata labeled by Ai65-tdtomato or rabies-dsRedXpress and catridges labeled by synaptotagmin-EGFP were automatically detected by a convolutional network trained as described previously (Ragan et al., 2012; Kim et al., 2017; Matho et al., 2021).

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
