## [Editor Report · eLife assessment]

The authors develop a novel genetic strategy for specific and comprehensive labeling of axo-axonic cells, also referred to as chandelier cells, in the mouse brain. The approach and analysis are rigorous such that the data **convincingly** support the key conclusions, including the expanded distribution of axo-axonic cells throughout the brain. This study provides **important** new information about the distribution of a significant neuronal cell type, as well as new tools for future studies. This work will be of broad interest to neuroscientists who work on the anatomical and functional organization of neural circuits.

---

## [Referee Report · Reviewer #2 (Public Review)]

Summary:

The goals of this study were to develop a genetic approach that would specifically and comprehensively target axo-axonic cells (AACs) throughout the brain and then to describe the patterns and characteristics of the targeted AACs in multiple, selected brain regions. The investigators have been successful in providing the most complete description of the regional distribution of putative (pAACs) throughout the brain to date. The supporting evidence is convincing, and the findings should serve as a guide for more detailed studies of AACs within each brain region and lead to new insights into their connectivity and functional organization of this important group of GABAergic interneurons.

Strengths:

The study has numerous strengths. A major strength is the development of a unique intersectional genetic strategy that uses cell lineage (Nkx2.1) and molecular (Unc5b or Pthlh) markers to identify AACs specifically and, apparently, nearly completely throughout the mouse brain. While AACs have been described previously in the cerebral cortex, hippocampus and amygdala, there has been no specific genetic marker that selectively identifies all AACs in these regions.

Importantly, the current genetic strategy labels pAACs in additional brain regions, including the claustrum-insular complex, extended amygdala, and several olfactory centers in which AACs have not been previously recognized. In general, the findings provide support for the specificity of the methods for targeting AACs and include several examples of labeling near markers of axon initial segments, providing validation of their AAC identity.

The descriptions and numerous low magnification images of the brain provide a roadmap for subsequent, detailed studies of AACs in numerous brain regions. The overview and summaries of the findings in the Abstract, Introduction and Discussion are particularly clear and helpful in placing the extensive regional descriptions of AACs in context.

Weaknesses:

Considering the unique and striking characteristics of AACs, it would have been ideal to include a clear, high resolution confocal image of an AAC from the Unc5b;Nkx2.1 mouse that would display the beauty of these cells with their numerous cartridges of axon terminals, emanating from a single AAC. While several cells are illustrated, the processes are often obscured by other labeling or the background created by the blue Dapi labeling. A high-resolution image of an isolated cell would not only support the identity of the cells as AACs but also demonstrate the potential advantages of their labeling for more detailed anatomical and neurophysiological studies. High magnification views of the axon terminals adjacent to AnkG-labeled axon initial segments are included and provide strong support for the identity of the cells. However, they cannot convey the extensiveness and patterns of the axonal arborizations of these cells.

The intersectional genetic methods included use of the lineage marker Nkx2.1 with either Unc5b or Pthlh as the molecular marker. As described, the mice with intersectional targeting of Nkx2.1 and Unc5b appear to show the most specific brain-wide labeling for AACs, and the majority of the descriptions are from these mice. The targeting with Nkx2.1 and Pthlh is less convincing and there appears to be a disconnect between the descriptions and the images. While the descriptions emphasize that the labeling is very similar in the two types of mice, the images suggest distinct differences, including labeling of non AACs in striatum and layer 4 of the cortex in the Pthlh;Nkx2.1 mouse, as described in the manuscript. In addition, the Pthlh;Nkx2.1 mouse has higher cell targeting in some regions and fewer labeled cells in others. Perhaps it would be more accurate to present the Pthlh;Nkx2.1 mouse as differing from the Unc5b;Nkx2.1 mouse, but useful for AAC labeling in select regions and under some conditions, such as following tamoxifen administration at specific ages. As currently presented, the inclusion of the Pthlh;Nkx2.1 detracts from the otherwise convincing argument that the Unc5b;Nkx2.1 mouse provides a specific and comprehensive way to identify AACs.

---

## [Referee Report · Reviewer #3 (Public Review)]

Summary:

Raudales et al. aimed at providing an insight into the brain-wide distribution and synaptic connectivity of bona fide GABAergic inhibitory interneuron subtypes focusing on the axo-axonic cell (AAC), one of the most distinctive interneuron subtypes, which innervates the axon initial segments of glutamatergic projection neurons. They establish intersectional genetic strategies that enable them to specifically and comprehensively capture AACs based on their lineage (Nkx2.1) and marker expression (Unc5b, Pthlh). They find that AACs are deployed across essentially all the pallium-derived brain structures as well as anterior olfactory nucleus, taenia tecta, and lateral septum. They show that AACs in distinct areas and layers of the neocortex as well as different subregions of the hippocampal formation display unique soma and synaptic density and morphological variations. Rabies virus-based retrograde monosynaptic input tracing reveals that AACs in the neocortex, the hippocampus, and the basolateral amygdala receive synaptic inputs from common as well as specific brain regions and supports the utility of this novel genetic approach. This study elucidates brain-wide neuroanatomical features and morphological variations of AACs with solid techniques and analysis. Their novel AAC-targeting strategies will facilitate the study of their development and function in different brain regions. The conclusions in this paper are well supported by the data. However, there are a few minor comments.

(1) The authors added a description about validation of ChCs in the method section: "Validation was conducted with high-magnification confocal microscopy and defined by a cell exhibiting at least two RFP-labelled axons colocalized with AIS labelled by AnkryinG or Phospho-IκBα". However, this does not clearly define pAACs themselves. If they follow this criteria, an RFP-labeled cell exhibiting only one synaptic cartridge that is colocalized with an AIS should be a pAAC. Is this what the authors are triying to say?

On the other hand, in the response to reviewers, the authors apparently define pAACs in a different way, in which they more focus on the number of cells exhibiting cartridges that are associated with AISs in a certain anatomical region rather than the number of cartridges per cell.

"For BNST we did not positively identify more than a few exhibiting overlap with AnkryinG/IκBα, so we currently leave them as pAACs"

"Putative AAC (pAACs) refers to populations in which relatively few single cell examples of AACs exhibiting co-localized cartridges were found"

The authors need to directly define pAACs.

(2) In the response to reviewers, the authors claimed that both Pthlh and Unc5b mice are useful for studying developing AACs. It would be nice if they include this content in the text (e.g. Discussion).

---

## [Author Response]

The following is the authors’ response to the original reviews.

**Public Reviews:**

**Reviewer #1 (Public Review):**
Summary:In this manuscript, the authors set out to develop genetic tools that can specifically and comprehensively label Axo-Axonic Cells (AACs), also known as Chandelier cells. These AACs possess unique morphological and connectivity features, making them an ideal subject for studying various aspects of cell types across different experimental methods. To achieve both specificity and comprehensiveness in AAC labeling, the authors employ an intersectional strategy that combines lineage origin and molecular markers. This approach successfully targets AACs across the mouse brain and reveals their widespread distribution in various brain structures beyond the previously known regions. Additionally, the authors utilize rabies transneuronal labeling to provide a comprehensive overview of AACs, their variations, and input sources throughout the brain. This experimental approach offers a powerful model system for investigating the role of AACs in circuit development and function across diverse brain regions.Strengths:Genetic Tools and Specificity: The authors' genetic tools show qualitative evidence of specificity for AACs, opening new avenues for targeted research on these cells. The use of intersectional strategies enhances the precision of AAC labeling.Widespread Distribution: The study significantly broadens our understanding of AAC distribution, revealing their presence in brain regions beyond what was previously documented. This expanded knowledge is a valuable contribution to the field.Transneuronal Labeling: The inclusion of rabies transneuronal labeling provides a comprehensive view of AACs, their variations, and input sources, allowing for a more holistic understanding of their role in neural circuits.Weaknesses:Quantitative Analysis: While the claim of specificity appears qualitatively convincing, the manuscript could be improved with more quantitative analysis.

We are glad that the reviewers appreciated our multimodal and brain-wide characterizations of the AAC population. We include many qualitative AAC examples and would like to highlight the quantitative nature of our whole brain cell body and cartridge analyses, made possible by transgenic targeting and our serial two-photon tomography imaging platform (STP). In addition to providing this brain wide AAC atlas, we also propose AACs as perhaps one of the best case examples for a bona fide cell type, which may inspire further in-depth anatomical and functional studies of AACs, and efforts to capture other ground truth cell types.

Comprehensiveness Claim: The assertion of comprehensiveness, implying labeling "almost all" AACs in all brain regions, is challenging to substantiate conclusively. Acknowledging the limitations of proving complete comprehensiveness and discussing them in the discussion section would be more appropriate than asserting it in the results section.

We thank the reviewer for this suggestion and have revised the results and discussion sections accordingly. The issue of how to access comprehensiveness in AAC labeling is a fair and important point, as dense brain-wide AAC labeling has not been achieved and assessed before. Previous studies had used less efficient and specific methods for capturing AACs, primarily in select areas of cortex, hippocampus, and amygdala. These AAC populations are recapitulated by our genetic strategies with higher density and specificity. It does not seem that we have missed any previously-reported AAC populations; in fact, we discovered multiple previously unreported populations. Another evidence supporting our “comprehensive” labeling of AACs is that two independent *Unc5b* and *Pthlh* transgenic strategies showed very similar AAC distribution patterns (Fig. 1 Suppl. 3). However, we recognize that probably the only way to fully assess “completeness” of labeling may be to compare with anatomical ground truth, such as by dense EM reconstruction of all AACs across the brain volume. This is currently not technically possible but may become feasible in the future.

Local Inputs: While the manuscript focuses on inter-areal inputs to AACs, it would benefit from exploring local inputs as well. Identifying the local neurons that target AACs and analyzing their patterns could provide valuable insights into AAC function within specific brain regions.

This is a good suggestion. However, our serial two-photon tomography imaging platform does not have the capability for reliably preserving tissue sections for immunohistochemical processing afterward. Additionally, though our starter AAV injections were limited to 100-150nL, there were far too many input cells labelled at the injection side to resolve individual input cells and correlate with their synaptic partners (e.g. a rabies-labelled pyramidal cell within the injection site may still project to starter cell few hundred microns away). Thus, our rabies input mapping was best suited for characterizing long-range inputs and was the focus here. For studying local inputs to AACs, future studies could combine very dilute starter AAV injections with multi-marker characterization of cell types by immunohistochemistry or FISH.

Discussion Focus: The discussion section should delve deeper into the biological implications of the findings, moving beyond technical significance. Exploring similarities and differences in input patterns between AACs and other cell types, and linking them to the locations of starter cells or specific connectivity patterns in the brain, would enrich the discussion. For instance, investigating whether input patterns can be predicted based on the locations of starter cells or connectivity specificity could provide valuable insights.

We thank the reviewer for this suggestion. We have expanded the discussion to include more on the relevance and implications of our input mapping results to different starter populations of AACs.

**Reviewer #2 (Public Review):**
Summary:The goals of this study were to develop a genetic approach that would specifically and comprehensively target axo-axonic cells (AACs) throughout the brain and then to describe the patterns and characteristics of the targeted AACs in multiple, selected brain regions. The investigators have been successful in providing the most complete description of the regional distribution of putative (pAACs) throughout the brain to date. The supporting evidence is convincing, even though incomplete in some brain regions. The findings should serve as a guide for more detailed studies of AACs within each brain region and lead to new insights into the connectivity and functional organization of this important group of GABAergic interneurons.Strengths:The study has numerous strengths. A major strength is the development of a unique intersectional genetic strategy that uses cell lineage (Nkx2.1) and molecular (Unc5b or Pthlh) markers to identify axo-axonic AACs specifically and, apparently, nearly completely throughout the mouse brain. While AACs have been described previously in the cerebral cortex, hippocampus, and amygdala, there has been no specific genetic marker that selectively identifies all AACs in these regions.The current genetic strategy has labeled pAACs in a large number of additional brain regions, including the claustrum-insular complex, extended amygdala, and several olfactory centers. In general, the findings provide support for the specificity of the methods for targeting AACs, and include some examples of labeling near markers of axon initial segments. However, the Investigators are careful to refer to labeled neurons as "putative AACs" as they have not been fully characterized and their identity verified.The descriptions and numerous low-magnification images of the brain provide a roadmap for subsequent, detailed studies of AACs in numerous brain regions. The overview and summaries of the findings in the Abstract, Introduction, and Discussion are particularly clear and helpful in placing the extensive regional descriptions of AACs in context.Weaknesses:One weakness of the study is the lack of an illustration of the high-resolution cell labeling that can be achieved with the methods, including labeling of numerous rows of axon terminals in contact with axon initial segments. The initial images of the brain-wide distribution of putative AACs are necessarily presented at low magnification. Although the authors indicate that the cells have "highly characteristic AAC labeling patterns throughout the neocortex, hippocampus and BLA", these morphological details cannot be visualized by the reader at the current magnification, even when the images are enlarged on the computer screen. Some of the details become evident in later Figures, but an initial illustration of single cell labeling with confocal microscopy, or tracing of their characteristic axonal arbors, would support the specificity of the labeling in the low magnification images.

We thank the reviewer for the suggestion. We have now added high-resolution images showing the colocalization of AAC axon boutons (cartridges) along AnkG positive postsynaptic axon initial segments in Fig. 2 Suppl. 1, Figure 1 panels a, d, e, and Fig. 4 panels b, c. These images unequivocally demonstrate AAC identity and specificity.

Table 1 indicates that the AAC identity of the cells has been validated in many brain regions but not in all. The methods used for validation have not been described and should be included for completeness. The authors are careful to acknowledge that labeled cells in some regions have not been validated and refer to such cells as pAACs.

Validation was defined by colocalization of RFP-labelled AAC cartridges and AnkryinG or Phospho-IκBα-labelled axon initial segments, imaged by confocal microscopy. We provide high-magnification examples throughout figures 2-6 and supplements. We have also tried to clarify this better in the methods section entitled “Immunohistochemistry.” Putative AAC (pAACs) refers to populations in which relatively few single cell examples of AACs exhibiting co-localized cartridges were found, largely due to the sparsity of the low tamoxifen dosage used (see response above).

The intersectional genetic methods included the use of the lineage marker Nkx2.1 with either Unc5b or Pthlh as the molecular marker. As described, the mice with intersectional targeting of Nkx2.1 and Unc5b appear to show the most specific brain-wide labeling for AACs, and the majority of the descriptions are from these mice. The targeting with Nkx2.1 and Pthlh is less convincing. The title for Figure 1 Supplemental Figure 3 suggests a similar AAC distribution in the Pthlh;Nkx2.1 mouse compared to the Unc5b;Nkx2.1 mouse. However, the descriptions of the individual panels suggest a number of inconsistencies and non-AAC labeling. The heavy labeling in the caudate and cells in layer 4 is particularly problematic. Based on the data presented, it appears that heavy labeling achieved in these mice could not be relied on for specific labeling of all AACs, although specific labeling could be achieved under some conditions, such as following tamoxifen administration at select ages.

The reviewer is correct about *Pthlh* being less specific for AACs than *Unc5b* when crossed to a constitutive *Nkx2.1* recombinase driver line. *Pthlh*/*Nkx2.1* intersection labeled a set of layer 4 cells in somatosensory cortex and dense cells in striatum, which are clearly not AACs. But these are the only main difference compared to *Unc5b*/*Nkx2.1* intersection. As the reviewer points out, it is only when *Pthlh* is crossed to an inducible *Nkx2.1-CreER* line and induced embryonically with tamoxifen that there is more specific AAC labeling (at least in cortex). We included this data as well as the intersection with *VIP-Cre* in case either of these are useful to researchers studying fate-mapping of AACs or bipolar cell interneurons. We have also revised the title of Fig. 1 Suppl. 3 to better convey this.

The methods described for dense labeling and single-cell labeling are described briefly in the methods. Some discussion of the development of the methods would be useful, including how it was determined that methods for heavy labeling identified AACs specifically and completely.

We have added a description on the development of these to the methods section entitled “Animals.”

**Reviewer #3 (Public Review):**
Summary:Raudales et al. aimed at providing an insight into the brain-wide distribution and synaptic connectivity of bona fide GABAergic inhibitory interneuron subtypes focusing on the axo-axonic cell (AAC), one of the most distinctive interneuron subtypes, which innervates the axon initial segments of glutamatergic projection neurons. They establish intersectional genetic strategies that enable them to specifically and comprehensively capture AACs based on their lineage (Nkx2.1) and marker expression (Unc5b, Pthlh). They find that AACs are deployed across essentially all the pallium-derived brain structures as well as the anterior olfactory nucleus, taenia tecta, and lateral septum. They show that AACs in distinct areas and layers of the neocortex as well as different subregions of the hippocampal formation display unique soma and synaptic density and morphological variations. Rabies virus-based retrograde monosynaptic input tracing reveals that AACs in the neocortex, the hippocampus, and the basolateral amygdala receive synaptic inputs from common as well as specific brain regions and supports the utility of this novel genetic approach. This study elucidates brain-wide neuroanatomical features and morphological variations of AACs with solid techniques and analysis. Their novel AAC-targeting strategies will facilitate the study of their development and function in different brain regions. The conclusions in this paper are well supported by the data. However, there are a few comments to strengthen this study.(1) The definition of putative AAC (pAAC) is unclear and Table 1 may not be accurate. Although the authors find synaptic cartridges of RFP-labeled cells in the claustro-insular complex and the dorsal endopiriform nuclei, they still consider these cells as pAACs (not validated). The authors claim that without examining the presence of synaptic cartridges, RFP-labeled cells in the hypothalamus and the bed nuclei of the stria terminalis (BNST) are pAACs while those in the L4 of the somatosensory cortex in Pthlh;Nkx2.1;Ai65 mice are non-AACs. In Table 1, the BNST is supposed to contain AACs (validated), but in the text, the authors claim that RFP-labeled cells in the BNST are pAACs. Could the authors clarify how AACs, pAACs, and non-AACs are defined?

We thank the reviewer for their interest and comments on our work. Please see our response to reviewer 2 for clarification on putative pAACs. Additionally, we have clarified in the methods under “Immunohistochemistry” how we defined AACs, pAAC, and non-AACs. For BNST we did not positively identify more than a few exhibiting overlap with AnkryinG/IκBα, so we currently leave them as pAACs—Table 1 has been corrected to reflect this.

(2) The intersectional strategies presented in this study could also specifically capture developing AACs. If so, how early are AACs labeled in the brain? It would also be nice if the authors could add a simple schematic like Fig. 1a showing the time course of Pthlh expression.

We thank the reviewer for suggesting the application of our method in studying AAC development. As the onset of *Unc5b* is in early postnatal time, tamoxifen induction of *Unc5b-CreER* in early postnatal days can enable studies of AAC neurite and synapse development, maturation, and plasticity. Similarly, *Pthlh* expression in the brain is relatively low/absent at P4 and present at P14 and later timepoints. *Pthlh-Flp;Nkx2.1-Cre* intersection can be used to study postnatal AAC development and plasticity.

**Recommendations for the authors:**

**Reviewer #1 (Recommendations For The Authors):**
While the claim of specificity appears qualitatively convincing, additional quantitative analysis would make the authors' claim much stronger. For example in Figure 4 (f-h), where the authors show an overlap of AAC axons with AnkG labeling, there also appears to be a region of AAC axon lacking adjacent AnkG labeling. The author could quantify the fraction of cartridges that overlap with AnkG labeling in different brain regions, potentially stringing their claim that pAACs are AACs as well as providing important documentation of the diversity or homogeneity of compartment targeting across the brain.

As mentioned previously, we only performed AnkG co-labeling analysis on low-dose tamoxifen/sparsely labelled samples in which we could readily differentiate individual cells. This was performed on samples with the Ai65 cytoplasmic reporter—for validation purposes we could positively identify co-labelled cartridges, but it would be more difficult to accurately identify any cartridges not co-labeled (since the entire axon was labelled with RFP). For precisely identifying and mapping AAC cartridge locations we found the intersectional synaptophysin-EGFP reporter (Fig. 2k-n) to be a more precise method for specifically labeling the “cartridge” segment of AAC axons. However, we did not try AnkG staining on samples from this reporter line, as they were set aside for STP imaging.

Regarding the claim of comprehensiveness, labeling "almost all" AACs in all brain regions is a high standard and challenging to demonstrate conclusively. The study already significantly expands our understanding of AAC distribution, and the authors might consider discussing the limitations of proving complete comprehensiveness in the discussion rather than claiming it in the results section.

We again thank the reviewer for this critique. As mentioned above, we have revised the results and discussion sections to better convey this point across.

Furthermore, the manuscript connectivity section primarily focuses on inter-areal inputs to AACs, but it could benefit from exploring local inputs as well. By identifying the local neurons that target AACs, the authors could ask if there is any general property or rule of the local projections to AACs across the brain, or at least within the cortex. Moreover, a clear indication of the injection site would be helpful, particularly in Figure 7, where there seems to be some discrepancy between the histograms and fluorescent images regarding local projections. The histograms of Figure 7, seem to indicate that the local projection to AACs is a small fraction of all the presynaptic neurons, however, the fluorescent image for the SSp seems to suggest otherwise with many fluorescent cells in the injected area.

We thank the reviewer for these comments. Regarding the local inputs in the rabies tracing datasets, it is a limitation (as mentioned above) of our STP platform’s inability to preserve tissue for immunohistochemistry labeling as well as our relatively dense starter cell labeling. Instead, our focus here was on long-range inputs (i.e. outside the ipsilateral ARA area of injection), which was simply not known for these AAC populations. We have revised the Figure 7 legend and added a description in the methods section to more clearly indicate that we only included long-range input projections in the Figure 7 histograms.

In the discussion, the authors should delve more into the biological implications of their findings rather than solely emphasizing the technical significance. They could explore the similarities and differences in input patterns between AACs and other cell types, potentially linking them to the locations of their starter cells or specific connectivity patterns in the brain. For example, the authors could check if the input patterns could be predicted from the projections to the layers where their starter cells are located (either from an Atlas like the Allen Connectivity Atlas, or from retrograde rabies injections in the same locations). Can the differences between the input patterns to PVC and AAC be predicted for their location versus some specificity of connections?

Thank you for the extensive comment. We address this point above, and have revised our discussion accordingly.

**Reviewer #2 (Recommendations For The Authors):**
The Figure legends vary in completeness and quality.(1) The legend for Figure 1 is very informative, and section e-g serves as a useful guide, as the legend includes the names of the brain regions related to the abbreviations and also indicates the specific panels that show the identified structures. Because of the large number of structures and the number of panels in each Figure, it would be ideal to follow the same pattern in the remaining figures.(2) Several edits are needed in the legend for Figure 1 Supplement Figure 1. The descriptions of a-f could be improved by providing general terms to describe the brain regions associated with the latter list of abbreviations (as has been done with the identification of the cerebral cortex, hippocampus, and olfactory centers and their related panels). One suggestion would be to write out insula, claustrum, and endopiriform prior to listing the abbreviations (AI, CLA, EP) (b-c) and adding amygdaloid complex and extended amygdala before the abbreviations (COA, BLA, MeA) (d-f) and (BST) (d).

We thank the reviewer, as the suggestion of further expanding the abbreviations is a good one. As such, we have revised/reorganized the anatomical abbreviations in the figure legends for Figure 1 Supplement Figures 1, 2, and 3.

Descriptions for Panels g-j require editing to link the appropriate panels and the descriptions. Panels for BSTpr appear to be g-h (rather than f-g) and i,j rather than h-i.

We have fixed this typo in the legend for Figure 1 Supplement Figure 1.

Descriptions for Panels k-n could be edited to include abbreviations for the identified brain regions. For example, include the abbreviation ARHP after arcuate nuclei and indicate panels m-n (rather than j-l); include PVP after paraventricular and indicate panel n (rather than m); include DMPH after dorsomedial nuclei and indicate k-m (rather than j-l).

Thank you for the suggestion. We have expanded the abbreviations in Figure 1 Supplement 1 accordingly.

**Reviewer #3 (Recommendations For The Authors):**
(1) Please clarify if tdTomato, EGFP (from helper AAVs), and RFP (from rabies virus) are native signals or IHC signals in legends.

We have added the descriptors “native” or “stained” to all figure legends containing fluorescent images.

(2) Fig. 4b and c: Please add insets of high-magnification images showing AAC boutons along AnkG-labeled AISs.

We have added these insets to Fig. 4b and c.

(3) Fig. 7S1: It appears that d and e are reversed. Judging from the positions of starter cells, d is for PV-Cre? Please make sure. It is also better to draw the laminar border in d and e.

The original genotype labels are correct for Fig. 7S1 d and e. We have added the laminar borders as suggested.

(4) Fig. 9b: Just for consistency, please label with the name of the helper AAV.

Added.

(5) Line 617: intragranular>>>infragranular?

Corrected, thank you.

(6) It may be unclear to some readers if the images in the figures are from confocal or STP. The authors may want to clarify that all images in the figures are generated by confocal microscopy in the method section.

We have clarified this better in the methods section, “Microcopy and image analysis.”

(7) The authors should clarify that STP was used to map input cells to the brain in the result section.

We have added this description in the results section.